# FASTopic: Pretrained Transformer is a Fast, Adaptive, Stable, and Transferable Topic Model

**Xiaobao Wu**[1]* **Thong Nguyen**[2] **Delvin Ce Zhang**[3] **William Yang Wang**[4] **Anh Tuan Luu**[1]*

[1]Nanyang Technological University     [2]National University of Singapore
[3]The Pennsylvania State University     [4]University of California, Santa Barbara

`xiaobao002@e.ntu.edu.sg`     `e0998147@u.nus.edu`     `delvin.ce.zhang@gmail.com`
`william@cs.ucsb.edu`     `anhtuan.luu@ntu.edu.sg`

## Abstract

Topic models have been evolving rapidly over the years, from conventional to recent neural models. However, existing topic models generally struggle with either effectiveness, efficiency, or stability, highly impeding their practical applications. In this paper, we propose FASTopic, a fast, adaptive, stable, and transferable topic model. FASTopic follows a new paradigm: Dual Semantic-relation Reconstruction (DSR). Instead of previous conventional, VAE-based, or clustering-based methods, DSR directly models the semantic relations among document embeddings from a pretrained Transformer and learnable topic and word embeddings. By reconstructing through these semantic relations, DSR discovers latent topics. This brings about a neat and efficient topic modeling framework. We further propose a novel Embedding Transport Plan (ETP) method. Rather than early straightforward approaches, ETP explicitly regularizes the semantic relations as optimal transport plans. This addresses the relation bias issue and thus leads to effective topic modeling. Extensive experiments on benchmark datasets demonstrate that our FASTopic shows superior effectiveness, efficiency, adaptivity, stability, and transferability, compared to state-of-the-art baselines across various scenarios. [1]

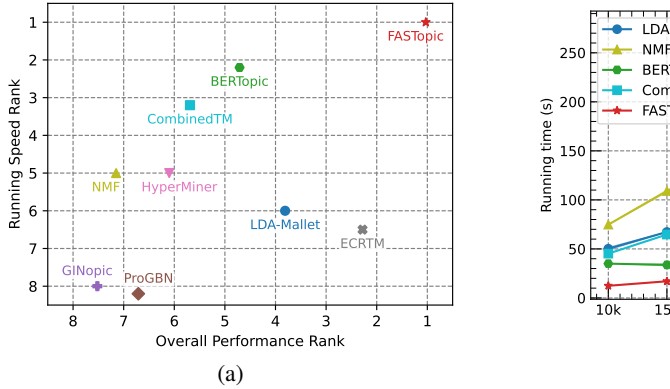

|     |     |
| :-: | :-: |
| (a) | (b) |

Figure 1: (**a**): Running speed rank and overall performance rank on the experiments with 6 benchmark datasets, including topic quality, doc-topic distribution quality, downstream tasks, and transferability. (**b**): Running time under the WoS dataset with varying sizes. See complete results in Figure 6.

---

*Corresponding authors.

[1]We release our code at `https://github.com/bobxwu/FASTopic` and our package at `https://pypi.org/project/fastopic/`.

38th Conference on Neural Information Processing Systems (NeurIPS 2024).

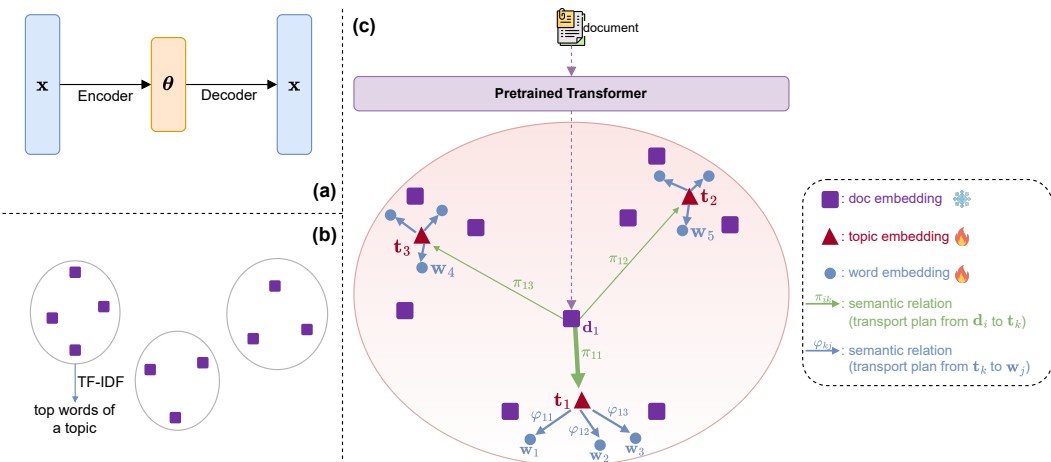

Figure 2: Illustration of topic modeling paradigms. **(a)**: VAE-based topic modeling with an encoder and a decoder [91, 65, 73]. **(b)**: Clustering-based topic modeling by clustering document embeddings [2, 24]. **(c)**: **Dual Semantic-relation Reconstruction** (**DSR**), modeling doc-topic distributions as the semantic relations between document (■) and topic embeddings (▲), and modeling topic-word distributions as the semantic relations between topic (▲) and word embeddings (●). Here we model these relations as the transport plans to alleviate the relation bias issue.

# 1   Introduction

Due to the unsupervised fashion and interpretability, topic models have derived a broad spectrum of applications [12, 14], such as content recommendation [39, 79], generation [17, 88], and trend analysis [18, 36]. Early conventional topic models follow probabilistic graphical models [10, 8] or non-negative matrix factorization [35, 58]. But they rely on laborious model-specific derivations and fall short on large-scale data [75]. Due to this, recent neural topic models have attracted more attention [90, 75], including VAE-based [42, 43, 61] and clustering-based [59, 89, 24].

However, existing neural topic models lack either efficiency, effectiveness, or stability. First, VAE-based topic models, while effective, are limited by their low efficiency. They follow the VAE framework [31] and often incorporate extra modules like graph neural networks [87, 1] or external knowledge [66, 80], resulting in complicated modeling structures. Owing to this, they suffer from intensive time complexity, *e.g.,* consuming hours to process a dataset of 10k documents [65]. Second, clustering-based topic models [89, 24] excel in efficiency as they require no training, but they sacrifice effectiveness. They tend to yield repetitive topics other than desired diverse ones or infer inaccurate topic distributions of documents [73, 1]. What is worse, these neural topic models suffer from low performance stability. They are extremely sensitive to hyperparameters, especially when applied to various scenarios concerning data domains, vocabulary sizes, and document length [27]. In consequence, these challenges hinder the applications of topic modeling in practice.

To tackle these challenges, we in this paper propose a **F**ast, **A**daptive, **S**table, and **T**ransferable topic model (**FASTopic**). Different from existing conventional, VAE-based, or clustering-based approaches, we introduce a new paradigm for topic modeling: **Dual Semantic-relation Reconstruction** (**DSR**) as illustrated in Figure 2. Instead of complicated neural networks, DSR only considers three parameters: document embeddings from a pretrained Transformer, and topic and word embeddings. DSR models the dual semantic relations between (1) document and topic embeddings, and (2) topic and word embeddings, and interprets them as distributions for topic modeling. By reconstruction with these relations, DSR discovers latent topics in a neat and efficient framework, avoiding the complicated structures in prior studies. To model these relations, we further propose the novel **Embedding Transport Plan** (**ETP**). Rather than simple parameterized softmax [37, 21], ETP regularizes these relations as the optimal transport plans between document, topic, and word embeddings. This mitigates the relation bias issue and produces distinct topics and accurate topic distributions, enabling effective topic modeling. Following the DSR paradigm with ETP, FASTopic provides a solid solution to the challenges of current topic models. As reported in Figure 1, FASTopic shows both superior efficiency and effectiveness compared to state-of-the-art baselines. Additionally FASTopic shows high

transferability, robust adaptivity and stability across various scenarios, delivering better performance without hyperparameter tuning. We conclude the main contributions of this paper as follows:

- We propose a novel topic model with a new dual semantic-relation reconstruction paradigm that models semantic relations among document, topic, and word embeddings, bringing about a neat and efficient topic modeling framework.
- We further propose a novel embedding transport plan method that regularizes the semantic relations as optimal transport plans, which avoids the relation bias issue and leads to effective topic modeling.
- We conduct extensive experiments and demonstrate that our model shows high effectiveness, efficiency, adaptivity, stability, and transferability compared to state-of-the-art baselines.

## 2 Related Work

**Conventional Topic Models**   These models have two types. The first type is probabilistic topic models [25, 7, 9, 8], *e.g.,* LDA [10], using probabilistic graphical models with topics as latent variables and inferred by Gibbs sampling [62] or Variational Inference [11]. The second type uses non-negative matrix factorization [29, 58]. These models have been extended to several scenarios like short texts [82, 67], multilingual [45], and dynamic topic modeling [9, 64]. But they require model-specific derivations for parameter inference and cannot well handle large-scale datasets.

**VAE-based Neural Topic Models**   These models follow the Variational AutoEncoder [VAE, 31, 55] framework and directly use gradient backpropagation to optimize parameters [42, 43, 61, 19, 83, 85, 84, 81, 47, 68–72, 76, 77, 74]. Although some work [91, 65, 86] like ECRTM [73] also uses optimal transport, we highlight that our method differs from them in that: (**i**) While they still follow the traditional complicated VAE framework, our FASTopic leverages the neat and efficient dual semantic-relation reconstruction paradigm; (**ii**) While they use optimal transport only as alternative distance measures or regularization for VAE, our FASTopic leverages the novel embedding transport plan to model semantic relations. These differences not only bring about faster running speed, but also lead to higher topic modeling performance.

**Clustering-based Neural Topic Models**   They cluster pretrained word embeddings via clustering algorithms like KMeans to yield topics [59, 2, 89], but mostly cannot infer topic distributions of documents. BERTopic [24] clusters the document embeddings and approximates the topic distributions by comparing documents to each document cluster. Different from simple clustering in these studies, we focus on explicitly modeling the complex relations among the embeddings of documents, topics, and words, which enhances topic modeling performance.

Some recent studies leverage large language models and describe topics as conceptual descriptions [53], rather than the word distributions in LDA [10]. They can reach higher interpretability, but we emphasize their two limitations: (i) They require more resources. They need to input each document as prompts to LLMs. This is time-consuming and computationally intensive, especially when handling large-scale datasets. (ii) They cannot produce precise distributions for topics and documents, which limits their applications in downstream tasks.

## 3 Methodology: FASTopic

In this section, we first recall the problem setting of topic modeling. Then we propose the new paradigm Dual Semantic-relation Reconstruction (DSR) and the novel Embedding Transport Plan (ETP) method. Finally we introduce our new **FASTopic**.

### 3.1 Problem Setting and Notations

Consider a collection $\{\mathbf{x}^{(1)}, \ldots, \mathbf{x}^{(N)}\}$ with $N$ documents and vocabulary size $V$. Topic modeling targets to discover $K$ latent topics from the collection. Following LDA [10], Topic#$k$ is defined as a distribution over all words, *i.e.,* topic-word distribution, denoted as $\boldsymbol{\beta}_k \in \mathbb{R}^V$. We have $\boldsymbol{\beta} = (\boldsymbol{\beta}_1, \ldots, \boldsymbol{\beta}_K) \in \mathbb{R}^{V \times K}$ as the topic-word distribution matrix of all topics. Topic modeling also infers the topic distributions of a document (what topics a document contains), *i.e.,* doc-topic distribution. We denote the doc-topic distribution of $\mathbf{x}^{(i)}$ as $\boldsymbol{\theta}^{(i)} \in \Delta_K$, with $\Delta_K$ as a probability simplex.

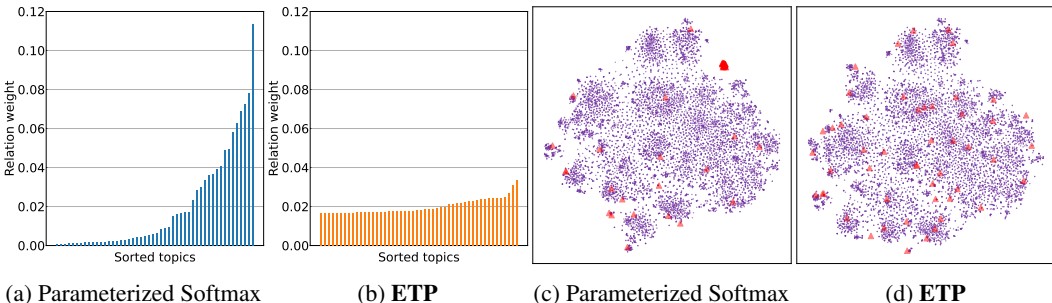

(a) Parameterized Softmax      (b) **ETP**      (c) Parameterized Softmax      (d) **ETP**

Figure 3: (**a, b**): Relation weights of topics to documents. (**c, d**): t-SNE visualization [63] of document (■), and topic (▲) embeddings under 50 topics ($K$=50). While most topic embeddings gather together in Parameterized Softmax (a,c) as it causes biased relations, ETP (b,d) separates all topic embeddings with regularized relations, avoiding the bias issue.

## 3.2   Dual Semantic-relation Reconstruction

In this section, we propose a new, neat, and efficient paradigm for topic modeling, **Dual Semantic-relation Reconstruction** (**DSR**). Figure 2 illustrates the differences between DSR and previous VAE-based and clustering-based methods.

**Parameterizing Documents, Topics, and Words**    At the beginning, we parameterize documents, topics, and words as embeddings. Specifically, we embed documents into an $H$-dimensional semantic space via a pretrained Transformer $f_{\text{doc}}$, *e.g.,* BERT [16] or Sentence-BERT [54]. Let $\mathbf{D}=(\mathbf{d}_1,\ldots,\mathbf{d}_N)\in\mathbb{R}^{H\times N}$ denote all document embeddings, where $\mathbf{d}_i=f_{\text{doc}}(\mathbf{x}^{(i)})$ refers to the embedding of $i$-th document. Then we randomly project all topics and words into the same semantic space as $K$ topic embeddings $\mathbf{T}=(\mathbf{t}_1,\ldots,\mathbf{t}_K)\in\mathbb{R}^{H\times K}$ and $V$ word embeddings $\mathbf{W}=(\mathbf{w}_1,\ldots,\mathbf{w}_V)\in\mathbb{R}^{H\times V}$. Notably we do *not* use pretrained word embeddings like word2vec [44] or GloVe [51], because they may not belong to the same semantic space as document embeddings, which hinders correctly measuring their distance.

**Reconstruction through Dual Semantic Relations**    Then we model the dual semantic relations between the embeddings of (1) documents and topics, and (2) topics and words. We interpret these relations as doc-topic distributions and topic-word distributions respectively. To be specific, we model $\theta_k^{(i)}$, the probability of Topic#$k$ given $i$-th document as the semantic relation between embeddings $\mathbf{d}_i$ and $\mathbf{t}_k$. Similarly, we model $\beta_{jk}$, the probability of $j$-th word given Topic#$k$ as the semantic relation between embeddings $\mathbf{t}_k$ and $\mathbf{w}_j$. We detail how to model them later in Sec. 3.3. We learn these relations through reconstruction. As shown in Figure 2, we reconstruct document $\mathbf{x}^{(i)}$ as $\boldsymbol{\beta}\boldsymbol{\theta}^{(i)}$, where we transport the semantics from $\mathbf{x}^{(i)}$ to each topic through $\boldsymbol{\theta}^{(i)}$ and then from each topic to each word through $\boldsymbol{\beta}$. Hence we formulate the objective for DSR as the reconstruction error:

$$\mathcal{L}_{\text{DSR}} = -\frac{1}{N}\sum_{i=1}^{N}(\mathbf{x}^{(i)})^{\top}\log(\boldsymbol{\beta}\boldsymbol{\theta}^{(i)}) \tag{1}$$

where we transform $\mathbf{x}^{(i)}$ into the Bag-of-Words following previous studies [42, 43, 61]. By minimizing this objective, we expect to push each topic embedding close to the embeddings of its semantically related documents and words; therefore we can learn informative semantic relations, *i.e.,* meaningful topic-word distributions $\boldsymbol{\beta}$ for latent topics and doc-topic distributions $\boldsymbol{\theta}^{(i)}$ for documents.

The above DSR presents a neat and efficient topic modeling paradigm. Previous VAE-based methods incorporate complicated modeling structures with diverse objectives [5, 80, 73, 1]. Different from them, DSR solely includes the objective Eq. (1) that only involves the embeddings of documents, topics, and words. This sufficiently simplifies the topic modeling procedure and hence facilitates efficiency. Moreover, while prior clustering-based methods [24] depend on indirect approximations, DSR explicitly models topic-word distributions and doc-topic distributions, resulting in higher effectiveness (See experiments in Sec. 4).

### 3.3 Embedding Transport Plan

In this section, we analyze how to model the semantic relations for topic modeling, and then propose a new solution **Embedding Transport Plan** (**ETP**).

**How to Model Semantic Relations?**   **We emphasize this is a *non-trivial* problem**. A common answer is the widely-used parameterized softmax function [28, 37, 21]:

$$\theta_k^{(i)} = \frac{\exp(-\|\mathbf{d}_i - \mathbf{t}_k\|^2/\tau)}{\sum_{k'=1}^{K} \exp(-\|\mathbf{d}_i - \mathbf{t}_{k'}\|^2/\tau)}, \quad \beta_{jk} = \frac{\exp(-\|\mathbf{t}_k - \mathbf{w}_j\|^2/\tau)}{\sum_{j'=1}^{V} \exp(-\|\mathbf{t}_k - \mathbf{w}_{j'}\|^2/\tau)} \tag{2}$$

where we measure the relation between embeddings as their Euclidean distance with hyperparameter $\tau$. Unfortunately, this straightforward way is ineffective, because it incurs the **relation bias issue**: most relations are minor as quantitatively illustrated in Figure 3a. In consequence, most topic embeddings fail to cover informative and distinct semantics but gather together in the space as shown in Figures 3c and 7a. This issue leads to repetitive topics and less accurate doc-topic distributions (See ablation studies in Sec. 4.7). Someone may guess this issue is because of the large topic number. We note that the relation bias issue still happens even under a small number of topics (See experimental supports in Table 9). Owing to these, we need alternatives to model the semantic relations.

**Transport Plan from Documents to Topics**   Motivated by the above analysis, we propose the new Embedding Transport Plan (ETP) to address the relation bias issue with effective regularization.

To regularize relations, we model them as the transport plan of a specifically defined optimal transport problem. In detail, we define two discrete measures $\gamma_1$ and $\rho_1$ over document and topic embeddings: $\gamma_1 = \sum_{i=1}^{N} \frac{1}{N} \delta_{\mathbf{d}_i}$ and $\rho_1 = \sum_{k=1}^{K} s_k \delta_{\mathbf{t}_k}$, where $\delta_x$ denotes the Dirac unit mass on $x$. We set the weight of each document embedding as $1/N$ and the weight of each topic embedding as $s_k$, where $\mathbf{s} = (s_1, \ldots, s_K)$ is a weight vector summing to 1. This later produces normalized doc-topic distributions. With these two, we formulate their entropic regularized optimal transport problem as

$$\underset{\boldsymbol{\pi} \in \mathbb{R}_+^{N \times K}}{\arg\min} \mathcal{L}_{\mathrm{OT}}(\gamma_1, \rho_1; \varepsilon_1) = \sum_{i=1}^{N} \sum_{k=1}^{K} C_{ik}^{(1)} \pi_{ik} + \varepsilon_1 \pi_{ik}(\log \pi_{ik} - 1), \text{ s.t. } \boldsymbol{\pi} \mathbb{1}_K = \frac{\mathbb{1}_N}{N} \text{ and } \boldsymbol{\pi}^\top \mathbb{1}_N = \mathbf{s}. \tag{3}$$

The first term is the original optimal transport problem, and the second term is the entropic regularization with hyperparameter $\varepsilon_1$ to make this problem tractable [13, 52]. This equation aims to find a transport plan $\boldsymbol{\pi}$ that minimizes the total cost of transporting the weights of document embeddings to topic embeddings under the two conditions [15], where $\mathbb{1}_K$ denotes a $K$-dimensional column vector of ones. Here $\pi_{ik}$ refers to the transport weight from $\mathbf{d}_i$ to $\mathbf{t}_k$. The transport cost between them is measured as Euclidean distance $C_{ik}^{(1)} = \|\mathbf{d}_i - \mathbf{t}_k\|^2$, with $\mathbf{C}^{(1)}$ as the transport cost matrix.

We can alleviate the relation bias issue with transport plan $\boldsymbol{\pi}$ as the semantic relations. Eq. (3) constraints $\boldsymbol{\pi}$ by two conditions. We set $\mathbf{s} = \mathrm{softmax}(\mathbf{s}_0)$, where $\mathbf{s}_0$ is a learnable variable uniformly initialized as $\frac{1}{K}\mathbb{1}_K$. The uniform initialization and softmax function prevent excessively biased $\mathbf{s}$ [28]. Therefore as illustrated in Figures 3b and 3d, this avoids biased $\boldsymbol{\pi}$ as it is constrained by $\mathbf{s}$, which mitigates the relation bias issue (See experiment results in Sec. 4.7). Besides, this approach flexibly captures the varying weight of each topic within the document collection, since some topics may appear more frequently in the collection while others less, which aligns with the reality (See more interpretations in Appendix E).

**Transport Plan from Topics to Words**   We further employ the transport plan between topic and word embeddings. We define two discrete measures over topic and word embeddings: $\gamma_2 = \sum_{k=1}^{K} \frac{1}{K} \delta_{\mathbf{t}_k}$ and $\rho_2 = \sum_{j=1}^{V} u_j \delta_{\mathbf{w}_j}$. Here we specify the weight of each topic embedding as $1/K$ and the weight of each word embedding as $u_j$, where $\mathbf{u}$ is a weight vector and its sum is 1. This is to produce normalized topic-word distributions later. Following Eq. (3), we write the entropic regularized optimal transport problem between the two measures as

$$\underset{\boldsymbol{\phi} \in \mathbb{R}_+^{K \times V}}{\arg\min} \mathcal{L}_{\mathrm{OT}}(\gamma_2, \rho_2; \varepsilon_2) = \sum_{k=1}^{K} \sum_{j=1}^{V} C_{kj}^{(2)} \phi_{kj} + \varepsilon \phi_{kj}(\log \phi_{kj} - 1), \text{ s.t. } \boldsymbol{\phi} \mathbb{1}_K = \frac{\mathbb{1}_K}{K} \text{ and } \boldsymbol{\phi}^\top \mathbb{1}_K = \mathbf{u}. \tag{4}$$

Table 1: Topic quality results of $C_V$ (topic coherence) and TD (topic diversity). The best is in **bold**. ‡ denotes the gain of FASTopic is statistically significant at 0.05 level.

| Model | 20NG | | NYT | | WoS | | NeurIPS | | ACL | | Wikitext-103 | |
|---|---|---|---|---|---|---|---|---|---|---|---|---|
| | $C_V$ | TD | $C_V$ | TD | $C_V$ | TD | $C_V$ | TD | $C_V$ | TD | $C_V$ | TD |
| LDA-Mallet | ‡0.371 | ‡0.763 | ‡0.362 | ‡0.747 | ‡0.352 | ‡0.866 | ‡0.366 | ‡0.573 | ‡0.357 | ‡0.570 | ‡0.407 | ‡0.607 |
| NMF | ‡0.372 | ‡0.473 | ‡0.378 | ‡0.403 | ‡0.388 | ‡0.430 | ‡0.375 | ‡0.292 | ‡0.362 | ‡0.300 | ‡0.411 | ‡0.429 |
| BERTopic | ‡0.406 | ‡0.654 | ‡0.413 | ‡0.679 | ‡0.437 | ‡0.649 | ‡0.379 | ‡0.472 | ‡0.399 | ‡0.462 | ‡0.432 | ‡0.608 |
| CombinedTM | ‡0.375 | ‡0.523 | ‡0.375 | ‡0.617 | ‡0.399 | ‡0.563 | ‡0.400 | ‡0.463 | ‡0.359 | ‡0.424 | ‡0.384 | ‡0.652 |
| GINopic | ‡0.369 | ‡0.862 | ‡0.381 | ‡0.701 | ‡0.362 | ‡0.576 | ‡0.392 | ‡0.452 | ‡0.370 | ‡0.537 | ‡0.385 | ‡0.699 |
| ProGBN | ‡0.378 | ‡0.587 | ‡0.381 | ‡0.574 | ‡0.371 | ‡0.806 | ‡0.379 | ‡0.363 | ‡0.384 | ‡0.521 | ‡0.406 | ‡0.617 |
| HyperMiner | ‡0.371 | ‡0.613 | ‡0.375 | ‡0.573 | ‡0.356 | ‡0.645 | ‡0.373 | ‡0.751 | ‡0.360 | ‡0.754 | ‡0.333 | ‡0.245 |
| ECRTM | **0.431** | ‡0.964 | 0.433 | 0.988 | ‡0.438 | **1.000** | 0.415 | ‡0.981 | ‡0.409 | ‡0.930 | ‡0.425 | ‡0.955 |
| **FASTopic** | 0.426 | **0.983** | **0.437** | **0.999** | **0.457** | **1.000** | **0.422** | **0.998** | **0.420** | **0.998** | **0.439** | **0.992** |

Here $\phi$ is the transport plan between topic embeddings and word embeddings, and $\phi_{kj}$ denotes the weight to be transported from topic embedding $\mathbf{t}_k$ to word embedding $\mathbf{w}_j$. We measure the transport cost as Euclidean distance $C_{kj}^{(2)} = \|\mathbf{t}_k - \mathbf{w}_j\|^2$. Similar to the above, we set $\mathbf{u} = \mathrm{softmax}(\mathbf{u}_0)$, where $\mathbf{u}_0$ is a learnable variable uniformly initialized as $\frac{1}{V}\mathbb{1}_V$. We model the semantic relations between topics and words with $\phi$ to mitigate the relation bias issue.

**Objective for ETP** With the above transport plans as semantic relations, we write $\boldsymbol{\theta}^{(i)}$ the doc-topic distribution of document $\mathbf{x}^{(i)}$ as

$$\boldsymbol{\theta}^{(i)} = N\boldsymbol{\pi}_i^*, \quad \text{where} \quad \boldsymbol{\pi}^* = \mathrm{sinkhorn}(\mathcal{L}_{\mathrm{OT}}(\gamma_1, \rho_1, \varepsilon_1)). \tag{5}$$

We employ Sinkhorn's algorithm [60, 15, 52] to compute the approximated solution $\boldsymbol{\pi}^*$ of the optimal transport problem in Eq. (3). As proved in early studies [57, 22, 23], $\boldsymbol{\pi}^*$ becomes a differentiable variable parameterized by transport cost matrix $\mathbf{C}^{(1)}$, which thus enables gradient backpropagation. See algorithm details in Appendix C. Here we rescale $\boldsymbol{\pi}_i^*$ by $N$ to output normalized $\boldsymbol{\theta}^{(i)}$ (the sum of each row in $\boldsymbol{\pi}^*$ is $1/N$ as previously constrained). In the same way, we model the topic-word distribution matrix $\boldsymbol{\beta}$ as

$$\boldsymbol{\beta} = K\boldsymbol{\phi}^*, \quad \text{where} \quad \boldsymbol{\phi}^* = \mathrm{sinkhorn}(\mathcal{L}_{\mathrm{OT}}(\gamma_2, \rho_2, \varepsilon_2)). \tag{6}$$

We rescale $\boldsymbol{\phi}^*$ by $K$ to produce normalized $\boldsymbol{\beta}$ (the sum of each row in $\boldsymbol{\phi}^*$ is $1/K$ as previously constrained). We formulate the objective for ETP by minimizing the total transport cost weighted by these approximated transport plans as

$$\mathcal{L}_{\mathrm{ETP}} = \sum_{i=1}^{N}\sum_{k=1}^{K} C_{ik}^{(1)}\pi_{ik}^* + \sum_{k=1}^{K}\sum_{j=1}^{V} C_{kj}^{(2)}\phi_{kj}^*. \tag{7}$$

This objective refines the embeddings by the regularized semantic relations, *i.e.,* the approximated transport plans under constraints.

### 3.4 Objective for FASTopic

Combining Eq. (1) and Eq. (7), we formulate the overall objective for FASTopic as

$$\min_{\mathbf{T},\mathbf{W},\mathbf{s},\mathbf{u}} \mathcal{L}_{\mathrm{ETP}} + \mathcal{L}_{\mathrm{DSR}}. \tag{8}$$

In short, $\mathcal{L}_{\mathrm{ETP}}$ refines topic and word embeddings with the regularized semantic relations; $\mathcal{L}_{\mathrm{DSR}}$ learns these relations by reconstruction. To reduce the number of hyperparameters, we set the weights of these two objectives as equal by default. See the training algorithm of FASTopic in Appendix C.

This objective is extremely simple compared to previous work with complex encoders and decoders following VAE [42, 61, 75]. It only optimizes four parameters: topic and word embeddings $\mathbf{T}, \mathbf{W}$, and their weights $\mathbf{s}$ and $\mathbf{u}$. We freeze document embeddings $\mathbf{D}$, because they have been pretrained already, and we may encounter over-fitting problems if we fine-tune them on a relatively small dataset. Due to this simple objective, our FASTopic enjoys super fast training. Previous models like ECRTM

Table 2: Text clustering results of Purity and NMI. The best is in **bold**.

| Model | 20NG | | NYT | | WoS | |
|---|---|---|---|---|---|---|
| | Purity | NMI | Purity | NMI | Purity | NMI |
| LDA-Mallet | ‡0.514 | ‡0.451 | ‡0.627 | ‡0.333 | ‡0.638 | ‡0.329 |
| NMF | ‡0.180 | ‡0.168 | ‡0.485 | ‡0.215 | ‡0.540 | ‡0.228 |
| BERTopic | ‡0.451 | ‡0.423 | ‡0.617 | ‡0.319 | ‡0.656 | ‡0.340 |
| CombinedTM | ‡0.459 | ‡0.435 | ‡0.600 | ‡0.323 | ‡0.655 | ‡0.346 |
| GINopic | ‡0.385 | ‡0.278 | ‡0.584 | ‡0.280 | ‡0.632 | ‡0.301 |
| ProGBN | ‡0.387 | ‡0.370 | ‡0.571 | ‡0.293 | ‡0.598 | ‡0.324 |
| HyperMiner | ‡0.433 | ‡0.405 | ‡0.579 | ‡0.315 | ‡0.636 | ‡0.349 |
| ECRTM | 0.560 | 0.524 | ‡0.615 | ‡0.357 | ‡0.650 | ‡0.349 |
| **FASTopic** | **0.577** | **0.525** | **0.662** | **0.369** | **0.672** | **0.365** |

Table 3: Running time (in seconds) on different datasets. The best is in **bold**.

| Model | 20NG | NYT | WoS | NeurIPS | ACL | Wikitext-103 |
|---|---|---|---|---|---|---|
| LDA-Mallet | 58.6 | 70.0 | 50.2 | 702.6 | 974.0 | 2083.0 |
| NMF | 87.0 | 81.4 | 74.8 | 268.6 | 399.4 | 939.0 |
| BERTopic | 34.2 | 35.2 | 35.0 | 55.6 | 66.8 | 114.2 |
| CombinedTM | 53.4 | 31.8 | 45.2 | 67.2 | 93.0 | 237.4 |
| GINopic | 417.8 | 334.2 | 309.2 | 905.8 | 664.8 | 1878.2 |
| ProGBN | 337.0 | 765.8 | 831.0 | 675.0 | 864.2 | 2180.2 |
| HyperMiner | 233.0 | 233.6 | 250.3 | 184.4 | 265.8 | 813.6 |
| ECRTM | 365.4 | 274.8 | 290.4 | 287.8 | 325.4 | 1270.0 |
| **FASTopic** | **10.6** | **12.5** | **12.4** | **18.3** | **60.3** | **50.5** |

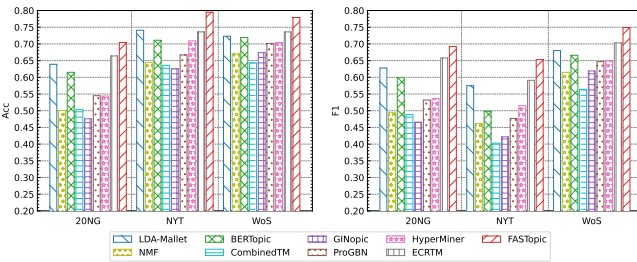

| Model | 20NG | | NYT | | WoS | |
|---|---|---|---|---|---|---|
| | Acc | F1 | Acc | F1 | Acc | F1 |
| LDA-Mallet | ‡0.481 | ‡0.472 | ‡0.699 | ‡0.502 | ‡0.629 | ‡0.526 |
| NMF | ‡0.328 | ‡0.317 | ‡0.626 | ‡0.414 | ‡0.526 | ‡0.418 |
| BERTopic | ‡0.299 | ‡0.285 | ‡0.550 | ‡0.361 | ‡0.514 | ‡0.416 |
| CombinedTM | ‡0.448 | ‡0.426 | ‡0.650 | ‡0.378 | ‡0.562 | ‡0.434 |
| GINopic | ‡0.087 | ‡0.082 | ‡0.314 | ‡0.099 | ‡0.330 | ‡0.110 |
| ProGBN | ‡0.178 | ‡0.169 | ‡0.504 | ‡0.280 | ‡0.443 | ‡0.273 |
| HyperMiner | ‡0.276 | ‡0.262 | ‡0.580 | ‡0.361 | ‡0.512 | ‡0.367 |
| ECRTM | ‡0.481 | ‡0.469 | ‡0.708 | ‡0.524 | ‡0.651 | ‡0.581 |
| **FASTopic** | **0.604** | **0.593** | **0.754** | **0.596** | **0.739** | **0.703** |

Figure 4: (**Left**): Text classification results of Accuracy (Acc) and F1. (**Right**): Transferability results. We use topic models trained on Wikitext-103 to infer the doc-topic distributions of other datasets. The best is in **bold**.

[73] also solve the optimal transport problem, but their objectives involve complicated encoders and decoders inherent from VAE, which slows them down.

Moreover, FASTopic needs much fewer hyperparameters. It mainly has hyperparameters for Sinkhorn's algorithm $\varepsilon_1$ and $\varepsilon_2$ in Eq. (3) and (4). VAE-based models, like CombinedTM [5], ECRTM [73], and GINopic [1], require hyperparameters to set their encoders, decoders (dimensions, number of layers, and dropout), and prior distributions (Gaussian or Dirichlet). BERTopic needs hyperparameters to set its clustering and dimension reduction modules, like the number of neighbors and components of UMAP; the min cluster size, min samples, metrics of HDBSCAN.

### 3.5 Inferring Doc-Topic distributions for New Documents

Finally we discuss how to infer doc-topic distributions for new documents. Considering a new document $\mathbf{x}'$ and its document embedding $\mathbf{d}'=f_{\text{doc}}(\mathbf{x}')$, we may directly follow the learning process in Sec. 3.3 and infer its doc-topic distribution $\boldsymbol{\theta}'$ by computing the transport plan between $\mathbf{d}'$ and learned topic embeddings $\mathbf{T}$. Unfortunately, this way is unworkable. It transports the weight of one document to all topics; hence for any $\mathbf{x}'$, the transport plan invariably becomes the learned topic weights $\mathbf{s}$. Such trivial results are certainly unreasonable. To this end, we compute $\boldsymbol{\theta}'$ as

$$\theta'_k = \frac{p_k}{\sum_{k'=1}^{K} p_{k'}}, \quad \text{where} \quad p_k = \frac{\exp(-\|\mathbf{t}_k - \mathbf{d}'\|^2/\tau)}{\sum_{i=1}^{N} \exp(-\|\mathbf{t}_k - \mathbf{d}_i\|^2/\tau)} \tag{9}$$

with $\tau$ as a temperature hyperparameter. Here we model the relation between $\mathbf{d}'$ and $\mathbf{t}_k$ as the Euclidean distance and regularize it by the total relations between $\mathbf{t}_k$ and all training documents to approximate the learned topic weight in Sec. 3.3. Then we compute $\theta'_k$ by normalizing over all topics. Since topic and word embeddings have been refined after training (Sec. 3.4), we can infer accurate doc-topic distributions for new documents in this way. See empirical results in Sec. 4.2 and 4.3.

## 4 Experiment

In this section we conduct comprehensive experiments to demonstrate that our FASTopic is fast, adaptive, stable, and transferable.

Table 4: Topic quality results of $C_V$ (topic coherence) and TD (topic diversity) under different topic numbers ($K$). The best is in **bold**.

| Model | K=75 | | K=100 | | K=125 | | K=150 | | K=175 | | K=200 | |
|---|---|---|---|---|---|---|---|---|---|---|---|---|
| | $C_V$ | TD | $C_V$ | TD | $C_V$ | TD | $C_V$ | TD | $C_V$ | TD | $C_V$ | TD |
| LDA-Mallet | ‡0.360 | ‡0.847 | ‡0.361 | ‡0.828 | ‡0.363 | ‡0.806 | ‡0.364 | ‡0.786 | ‡0.369 | ‡0.781 | ‡0.371 | ‡0.755 |
| NMF | ‡0.393 | ‡0.415 | ‡0.390 | ‡0.382 | ‡0.392 | ‡0.357 | ‡0.390 | ‡0.332 | ‡0.391 | ‡0.307 | ‡0.393 | ‡0.298 |
| BERTopic | ‡0.444 | ‡0.638 | ‡0.450 | ‡0.607 | 0.458 | ‡0.566 | 0.457 | ‡0.529 | 0.457 | ‡0.543 | 0.455 | ‡0.541 |
| CombinedTM | ‡0.388 | ‡0.482 | ‡0.385 | ‡0.473 | ‡0.388 | ‡0.455 | ‡0.387 | ‡0.428 | ‡0.390 | ‡0.425 | ‡0.389 | ‡0.443 |
| GINopic | ‡0.369 | ‡0.485 | ‡0.372 | ‡0.445 | ‡0.369 | ‡0.432 | ‡0.372 | ‡0.445 | ‡0.377 | ‡0.446 | ‡0.375 | ‡0.447 |
| ProGBN | ‡0.380 | ‡0.732 | ‡0.380 | ‡0.673 | ‡0.380 | ‡0.599 | ‡0.379 | ‡0.556 | ‡0.374 | ‡0.472 | ‡0.375 | ‡0.444 |
| HyperMiner | ‡0.356 | ‡0.586 | ‡0.350 | ‡0.595 | ‡0.351 | ‡0.568 | ‡0.352 | ‡0.547 | ‡0.349 | ‡0.518 | ‡0.355 | ‡0.490 |
| ECRTM | **0.482** | ‡0.974 | **0.477** | ‡0.951 | **0.461** | ‡0.958 | 0.458 | ‡0.957 | 0.448 | 0.953 | ‡0.437 | 0.941 |
| **FASTopic** | 0.465 | **0.998** | 0.464 | **0.993** | 0.460 | **0.984** | **0.459** | **0.972** | **0.458** | **0.959** | **0.456** | **0.949** |

Table 5: Document clustering results of Purity and NMI under different topic numbers ($K$). The best is in **bold**.

| Model | K=75 | | K=100 | | K=125 | | K=150 | | K=175 | | K=200 | |
|---|---|---|---|---|---|---|---|---|---|---|---|---|
| | Purity | NMI | Purity | NMI | Purity | NMI | Purity | NMI | Purity | NMI | Purity | NMI |
| LDA-Mallet | ‡0.661 | ‡0.333 | ‡0.651 | ‡0.322 | ‡0.660 | ‡0.325 | ‡0.675 | ‡0.332 | ‡0.684 | ‡0.332 | ‡0.676 | ‡0.327 |
| NMF | ‡0.570 | ‡0.245 | ‡0.582 | ‡0.256 | ‡0.593 | ‡0.263 | ‡0.606 | ‡0.272 | ‡0.604 | ‡0.270 | ‡0.613 | ‡0.273 |
| BERTopic | ‡0.672 | ‡0.333 | 0.696 | ‡0.340 | 0.704 | ‡0.338 | 0.713 | ‡0.338 | ‡0.720 | ‡0.343 | ‡0.722 | ‡0.347 |
| CombinedTM | ‡0.672 | ‡0.343 | ‡0.683 | ‡0.347 | ‡0.695 | ‡0.348 | ‡0.705 | ‡0.352 | ‡0.710 | ‡0.355 | ‡0.709 | ‡0.350 |
| GINopic | ‡0.644 | ‡0.308 | ‡0.654 | ‡0.311 | ‡0.662 | ‡0.319 | ‡0.672 | ‡0.318 | ‡0.682 | ‡0.323 | ‡0.687 | ‡0.327 |
| ProGBN | ‡0.624 | ‡0.313 | ‡0.641 | ‡0.315 | ‡0.627 | ‡0.303 | ‡0.637 | ‡0.302 | ‡0.664 | ‡0.320 | ‡0.663 | ‡0.316 |
| HyperMiner | ‡0.641 | ‡0.339 | ‡0.655 | ‡0.342 | ‡0.661 | ‡0.331 | ‡0.664 | ‡0.330 | ‡0.654 | ‡0.320 | ‡0.666 | ‡0.324 |
| ECRTM | ‡0.648 | ‡0.334 | ‡0.668 | ‡0.343 | ‡0.680 | ‡0.343 | ‡0.697 | ‡0.356 | ‡0.700 | ‡0.355 | ‡0.701 | 0.361 |
| **FASTopic** | **0.689** | **0.364** | **0.703** | **0.368** | **0.710** | **0.365** | **0.721** | **0.368** | **0.735** | **0.372** | **0.735** | **0.368** |

## 4.1 Experiment Setup

**Datasets** We adopt six benchmark datasets for experiments: (**i**) **20NG** [20Newsgroup, 33] is one of the most commonly-used datasets, covering news articles with 20 labels. (**ii**) **NYT** includes news articles from the New York Times with 12 categories. (**iii**) **WoS** [Web Of Science, 32] contains published papers from the Web of Science website with 7 categories. (**iv**) **NeurIPS** is a dataset with papers published at the NeurIPS conference from 1987 to 2017. (**v**) **ACL** [6] contains research articles from the ACL anthology from 1970 to 2015. (**vi**) **Wikitext-103** [41] includes Wikipedia articles. See more dataset details in Appendix B.

**Evaluation Metrics** Although topic modeling evaluation is still an open problem [26], we follow mainstream studies [19, 91, 73] and evaluate the topic quality and doc-topic distribution quality. For topic quality, we consider: (**i**) **Topic Coherence** measures the coherence between top words of discovered topics. We employ the widely-used coherence metric $C_V$, which has been shown to outperform earlier NPMI, UCI, and UMass [46, 34, 56]. We use a widely-used large Wikipedia article collection as the external reference corpus to compute $C_V$. (**ii**) **Topic Diversity** means the differences between discovered topics. We measure this with the Topic Diversity (TD) metric [19], which calculates the proportion of unique words in the topics. In terms of doc-topic distribution quality, we conduct document clustering, evaluated by Purity and NMI [38] following Zhao et al. [91]. We do not evaluate the perplexity since our method does *not* follow the VAE framework [42, 43], and the perplexity is incomparable across topic models as evidenced by early studies [90, 75].

**Baseline Models** We consider the following baselines in three paradigms. For conventional topic models, we adopt (**i**) **LDA-Mallet** [40], a prominent method competitive to some neural models [26]; (**ii**) **NMF**, using non-negative matrix factorization. For clustering-based topic models, we have (**iii**) **BERTopic** [24], clustering document embeddings and discovering topics by TF-IDF. For VAE-based neural topic models, we include (**iv**) **CombinedTM** [5], combining contextual features and BoW as inputs; (**v**) **GINopic** [1], following CombinedTM but using graph isomorphism networks; (**vi**) **HyperMiner** [80], using hyperbolic embeddings to model topics; (**vii**) **ProGBN** [20], progressively generating documents of different levels with graph decoders; (**viii**) **ECRTM** [73],

a state-of-the-art method by regularizing embeddings with optimal transport. We fine-tune the hyperparameters of these baselines under different datasets and topic numbers.

## 4.2 Effectiveness: Topic Quality and Doc-Topic Distribution Quality

We demonstrate the superior effectiveness of FASTopic compared to state-of-the-art baselines. Table 1 presents the topic quality results of topic coherence ($C_V$) and topic diversity (TD). We see that our FASTopic commonly surpasses all baselines with the highest performance across all datasets. Moreover, Table 2 reports the doc-topic distribution quality results concerning the Purity and NMI of document clustering. We observe that FASTopic reaches top performance as well. These results manifest that FASTopic produces high-quality topics and doc-topic distributions, showing better effectiveness. This also verifies the capability of our new DSR paradigm. See Appendix H for the examples of discovered topics.

## 4.3 Effectiveness: Text Classification as Downstream Task

We consider text classification as a downstream task to evaluate topic models in an extrinsic manner. Following Wu et al. [73], we train SVM classifiers with the inferred doc-topic distributions as document features and then predict the class of each testing document. We measure this performance by Accuracy (Acc) and F1. Figure 4 reports that our FASTopic consistently outperforms baselines. We note that the improvements of FASTopic are statistically significant at 0.01 level. These results demonstrate that FASTopic can benefit more downstream classification tasks.

## 4.4 Efficiency: Running Speed

We show the exceptionally fast running speed of FASTopic. Table 3 reports the running time of each model on each dataset. The running time indicates the duration from the completion of data loading to the finish of training. We see that our FASTopic consistently emerges as the fastest one by a large margin, statistically significant at 0.01 level. FASTopic completes running within 1 minute, while the longest takes 30 minutes. We notice that LDA-Mallet has increasing running time on the datasets with longer documents. For instance, it escalates from 50 seconds on 20NG to 2000 seconds on Wikitext-103. In contrast, FASTopic maintains its rapid performance regardless of document length. Figure 1b also evidences the fast speed of FASTopic in terms of varying dataset sizes. This is because FASTpoic adopts our neat and efficient DSR paradigm, which relieves from the complicated modeling structures in previous studies. See more running time analysis in Appendix G.

## 4.5 Transferability

We verify the high transferability of FASTopic. In detail, we train a topic model on Wikitext-103, a general dataset with diverse topics, and then use it to infer the doc-topic distributions of documents in other datasets 20NG, NYT, and WoS. Following the previous setting, we use these doc-topic distributions as features to train SVM classifiers for text classification. This measures the transferability of a topic model from one data domain to another. Figure 4 shows that the transferability of FASTopic significantly outperforms baselines. The reason lies in that previous methods often rely on the Bag-of-Words [80, 20, 73]. Differently, FASTopic leverages richer representations, the pretrained document embeddings, and learns the doc-topic distributions through the effective ETP method, bringing about higher transferability.

## 4.6 Adaptivity and Stability

We demonstrate the adaptivity and stability of FASTopic across various scenarios using the WoS dataset. First, Tables 4 and 5 summarize the performance under different topic numbers ($K$ from 75 to 200). We observe that FASTopic generally remains top performance across these variations. Second, Tables 13 and 14 report the results under varying dataset sizes ($N$ from 15k to 40k). These results show that FASTopic mostly reaches the best results. As aforementioned in Figure 1b, FASTopic also has the fastest running speed. Third, we experiment with different vocabulary sizes ($V$ from 20k to 50k) in Tables 15 and 16. Similarly, our FASTopic exhibits stable and high performance. We note that FASTopic uses the same hyperparameters in all these experiments (See Appendix D). The above

Table 6: Ablation study. w/o ETP means using parameterized softmax (Eq. (2)) to model semantic relations. See also Table 8 for results on other datasets.

| Model | 20NG | | | | NYT | | | | WoS | | | |
|---|---|---|---|---|---|---|---|---|---|---|---|---|
| | $C_V$ | TD | Purity | NMI | $C_V$ | TD | Purity | NMI | $C_V$ | TD | Purity | NMI |
| w/o ETP | ‡0.368 | ‡0.391 | ‡0.401 | ‡0.452 | ‡0.363 | ‡0.338 | ‡0.553 | ‡0.294 | ‡0.395 | ‡0.522 | ‡0.653 | ‡0.350 |
| **FASTopic** | 0.426 | 0.983 | 0.577 | 0.525 | 0.437 | 0.999 | 0.662 | 0.369 | 0.457 | 1.000 | 0.672 | 0.365 |

results together highlight that our FASTopic can smoothly adapt to various scenarios with stable performance. This is a vital advantage of our model for practical applications.

## 4.7 Ablation Study

We validate the necessity of our Embedding Transport Plan (ETP) method with ablation studies. Table 6 shows that using parameterized softmax rather than ETP (w/o ETP) to model semantic relations incurs degraded performance, concerning both topic and doc-topic distribution quality (See also the results on other datasets in Table 8). For instance, the $C_V$ and TD decrease from 0.426, 0.983 to 0.368, 0.391; the Purity and NMI decrease from 0.577, 0.525 to 0.401, 0.452, indicating low-quality repetitive topics and less accurate doc-topic distributions. We observe similar results even with only 10 topics ($K$=10) in Table 9. This is because our ETP properly regularizes semantic relations, which addresses the relation bias issue. These results manifest the necessity of our ETP to reach effective topic modeling.

## 5 Model Usage

We have released our FASTopic as a Python package at PyPI [2]. Users can easily install FASTopic through *pip*. Figure 5 shows a code example to use FASTopic on a dataset. After preprocessing the given dataset, it discovers top words and infers doc-topic distributions. With these simple APIs, users can smoothly handle their data for their various purposes. See our GitHub [3] for more tutorials and documentation of FASTopic.

```
#! pip install fastopic
from fastopic import FASTopic
from topmost.preprocessing import Preprocessing

# Preprocess the dataset.
docs = [ "doc 1", "doc 2", ...]
preprocessing = Preprocessing(stopwords="English")

model = FASTopic(num_topics=50, preprocessing)
topic_top_words, doc_topic_dist = model.fit_transform(docs)
```

Figure 5: A code example of using FASTopic. Install FASTopic via *pip* and use its APIs to handle a dataset.

## 6 Conclusion

In this paper, we propose FASTopic, a fast, adaptive, stable, and transferable topic model. Rather than traditional VAE-based or clustering-based approaches, FASTopic employs the new dual semantic-relation reconstruction paradigm to model latent topics with semantic relations and uses the new transport plan relation method to tackle the relation bias issue. Comprehensive experiments demonstrate the significantly superior performance of FASTopic in terms of effectiveness, efficiency, adaptivity, stability, and transferability. These advantages manifest the strong capability of FASTopic in practice, which benefits a wide range of real-world applications.

---

[2] https://pypi.org/project/fastopic/.
[3] https://github.com/bobxwu/FASTopic

## Acknowledgements

This research/project is supported by the National Research Foundation, Singapore under its AI Singapore Programme (AISG Award No: AISG2-TC-2022-005).

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

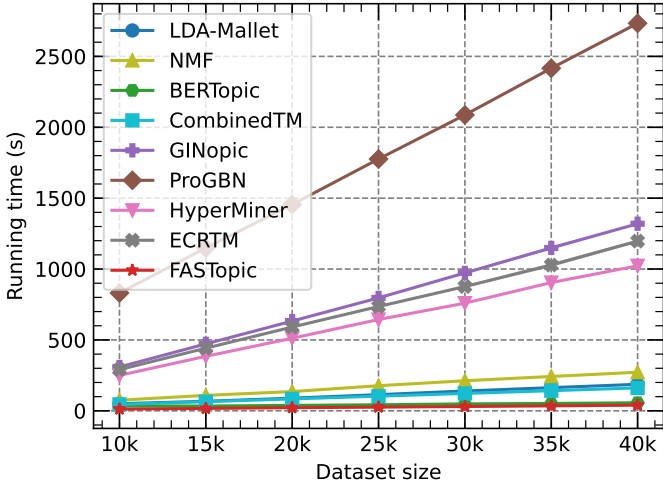

Figure 6: Running time under WoS with different data sizes. See also a zoomed-in view in Figure 1b.

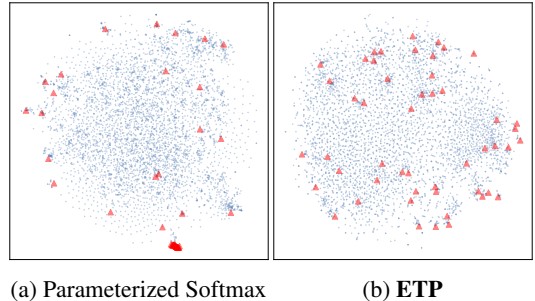

(a) Parameterized Softmax  (b) **ETP**

Figure 7: t-SNE visualization of topic (▲) and word (•) embeddings under 50 topics ($K$=50).

Table 7: Dataset statistics.

| Dataset | #docs | Average Length | Vocabulary size | #labels |
|---|---|---|---|---|
| 20NG | 18,846 | 110.5 | 5,000 | 20 |
| NYT | 9,172 | 175.4 | 10,000 | 12 |
| WoS | 10,000 | 110.0 | 10,000 | 7 |
| NeurIPS | 7,237 | 2,085.9 | 10,000 | — |
| ACL | 10,560 | 2,023.0 | 10,000 | — |
| Wikitext-103 | 28,532 | 1,355.4 | 10,000 | — |

# A  Limitations

Our method achieves promising performance, but we mention that one limitation is the max input length of pretrained document embedding models. This may hamper the performance on extremely long documents. However, we show that our method can well handle long documents like academic papers in Sec. 4, and this issue can be well resolved by newer and stronger pretrained models like large language models [48, 50, 78, 49].

**Algorithm 1** Training algorithm for FASTopic.

**Input:** document collection $\{\mathbf{x}^{(1)}, \mathbf{x}^{(2)}, \ldots, \mathbf{x}^{(N)}\}$, pretrained document embedding model $f_{\text{doc}}$;
**Output:** model parameters $\mathbf{T}, \mathbf{W}, \mathbf{s}, \mathbf{u}$;

1: `// Function of Embedding Transport Plan (ETP)`
2: **function** ETP($\mathbf{C}, \mathbf{v}, \varepsilon, L_1, L_2$)
3:     `// Sinkhorn's algorithm`;
4:     $\mathbf{M} = \exp(-\mathbf{C}/\varepsilon)$;
5:     $\mathbf{b} \leftarrow \mathbb{1}_{L_2}$;
6:     **while** not converged and not reach max iterations **do**
7:         $\mathbf{a} \leftarrow \frac{1}{L_1} \frac{\mathbb{1}_{L_1}}{\mathbf{Mb}}, \quad \mathbf{b} \leftarrow \frac{\mathbf{v}}{\mathbf{M}^\top \mathbf{a}}$;
8:     **end while**
9:     **return** $\text{diag}(\mathbf{a})\mathbf{M}\,\text{diag}(\mathbf{b})$;
10: **end function**

11: Initialize $\mathbf{s}_0 \leftarrow \frac{1}{K}\mathbb{1}_K$;
12: Initialize $\mathbf{u}_0 \leftarrow \frac{1}{V}\mathbb{1}_V$;
13: Initialize $\mathbf{d}_i \leftarrow f_{\text{doc}}(\mathbf{x}_i)$, for $i = \{1, 2, \ldots, N\}$;   `// Initialize D`
14: Randomly initialize $\mathbf{T}$ and $\mathbf{W}$;

15: **for** 1 to $n_{\text{epoch}}$ **do**
16:     $\mathbf{s} = \text{softmax}(\mathbf{s}_0)$
17:     $\mathbf{u} = \text{softmax}(\mathbf{u}_0)$
18:     `// Embedding transport between documents and topics`;
19:     $C_{ik}^{(1)} = \|\mathbf{d}_i - \mathbf{t}_k\|^2$;   `// Transport cost matrix between documents and topics`
20:     $\boldsymbol{\pi}^* = \text{ETP}(\mathbf{C}^{(1)}, \mathbf{s}, \varepsilon_1, N, K)$;   `// Transport plan between documents and topics`

21:     `// Embedding transport between topics and words`;
22:     $C_{kj}^{(2)} = \|\mathbf{t}_k - \mathbf{w}_j\|^2$;   `// Transport cost matrix between topics and words`
23:     $\boldsymbol{\phi}^* = \text{ETP}(\mathbf{C}^{(2)}, \mathbf{u}, \varepsilon_2, K, V,)$;   `// Transport plan between topics and words`

24:     $\boldsymbol{\beta} = K\boldsymbol{\phi}^*$
25:     $\boldsymbol{\theta}^{(i)} = N\boldsymbol{\pi}_i^*$, for $i = \{1, 2, \ldots, N\}$
26:     Compute Eq. (8);
27:     Update $\mathbf{T}, \mathbf{W}, \mathbf{s}, \mathbf{u}$ with a gradient step;
28: **end for**

Table 8: Ablation study. w/o ETP means using parameterized softmax (Eq. (2)) to model semantic relations.

| Model | NeurIPS | | ACL | | Wikitext-103 | |
|---|---|---|---|---|---|---|
| | $C_V$ | TD | $C_V$ | TD | $C_V$ | TD |
| w/o ETP | ‡0.380 | ‡0.216 | ‡0.373 | ‡0.164 | ‡0.359 | ‡0.335 |
| **FASTopic** | 0.422 | 0.998 | 0.420 | 0.998 | 0.439 | 0.992 |

# B   Preprocessing Datasets

We follow the dataset preprocessing steps of TopMost [76] [4]: (1) tokenize documents and convert to lowercase; (2) remove punctuation; (3) remove tokens that include numbers; (4) remove tokens less than 3 characters; (5) remove stopwords.

Table 7 reports the statistics of preprocessed datasets.

---

[4] `https://github.com/bobxwu/topmost`

Table 9: Ablation study under $K$=10. w/o ETP means using parameterized softmax (Eq. (2)) to model semantic relations.

| Model | 20NG | | | | NYT | | | | WoS | | | | NeurIPS | | ACL | | Wikitext-103 | |
|---|---|---|---|---|---|---|---|---|---|---|---|---|---|---|---|---|---|---|
| | $C_V$ | TD | Purity | NMI | $C_V$ | TD | Purity | NMI | $C_V$ | TD | Purity | NMI | $C_V$ | TD | $C_V$ | TD | $C_V$ | TD |
| w/o ETP | ‡0.367 | ‡0.571 | ‡0.294 | ‡0.408 | ‡0.371 | ‡0.620 | ‡0.512 | ‡0.263 | ‡0.407 | ‡0.537 | ‡0.603 | ‡0.381 | ‡0.379 | ‡0.512 | ‡0.364 | ‡0.412 | ‡0.416 | ‡0.660 |
| **FASTopic** | 0.432 | 1.000 | 0.331 | 0.437 | 0.428 | 1.000 | 0.549 | 0.316 | 0.432 | 1.000 | 0.629 | 0.414 | 0.434 | 1.000 | 0.442 | 1.000 | 0.469 | 1.000 |

Table 10: Topic quality results with different document embedding models.

| Doc Embedding Model | 20NG | | NYT | | WoS | | NeurIPS | | ACL | | Wikitext-103 | |
|---|---|---|---|---|---|---|---|---|---|---|---|---|
| | $C_V$ | TD | $C_V$ | TD | $C_V$ | TD | $C_V$ | TD | $C_V$ | TD | $C_V$ | TD |
| `all-mpnet-base-v2` | 0.424 | 0.986 | 0.427 | 0.999 | 0.464 | 1.000 | 0.415 | 0.996 | 0.426 | 0.996 | 0.435 | 0.997 |
| `all-distilroberta-v1` | 0.517 | 0.976 | 0.426 | 1.000 | 0.460 | 1.000 | 0.420 | 0.998 | 0.413 | 0.999 | 0.435 | 0.997 |
| `all-MiniLM-L6-v2` | 0.412 | 0.981 | 0.437 | 0.999 | 0.452 | 1.000 | 0.420 | 0.996 | 0.413 | 0.997 | 0.439 | 0.992 |

Table 11: Document clustering results with different document embedding models.

| Doc Embedding Model | 20NG | | NYT | | WoS | |
|---|---|---|---|---|---|---|
| | Purity | NMI | Purity | NMI | Purity | NMI |
| `all-mpnet-base-v2` | 0.611 | 0.550 | 0.669 | 0.381 | 0.680 | 0.368 |
| `all-distilroberta-v1` | 0.581 | 0.523 | 0.671 | 0.371 | 0.679 | 0.370 |
| `all-MiniLM-L6-v2` | 0.570 | 0.522 | 0.662 | 0.369 | 0.669 | 0.362 |

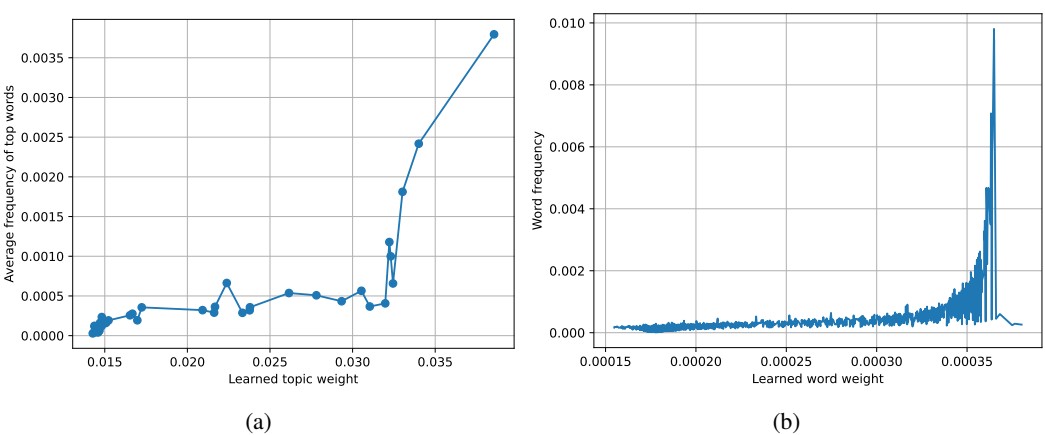

(a)                                    (b)

Figure 8: **(a)**: Learned topic weights and the average frequency of the top words in topics. **(b)**: Learned word weights and the word frequency.

## C  Training Algorithm for FASTopic

Algorithm 1 shows the training algorithm of FASTopic. ETP uses Sinkhorn's algorithm [60, 15] to compute approximated transport plan $\pi^*$ and $\phi^*$. It is iterative and fast, especially suited to the execution of GPU [23].

## D  Model Implementation

See our code for implementation details.

We conduct our experiments with A6000 GPU. We use `all-MiniLM-L6-v2` in Sentence-Transformers [5] to obtain the pretrained document embeddings, as it is fast and also offers good quality. See results with other embedding models in Appendix F. We set the maximum number of

---

[5] `https://huggingface.co/sentence-transformers`

| BERTopic | |
|---|---|
| Step 1: Load doc embeddings | 7.10 |
| Step 2: Reduce dimensionality | 23.13 |
| Step 3: Cluster doc embeddings | 0.21 |
| Step 4: Compute word weights | 1.97 |
| **Sum** | **32.42** |

(a)

| FASTopic | |
|---|---|
| Step 1: Load doc embeddings | 7.10 |
| Step 2: Training | 5.85 |
| **Sum** | **12.95** |

(b)

Table 12: Running time breakdowns (in seconds) of BERTopic and our FASTopic on the NYT dataset.

iterations as 1,000 and the stop tolerance as 0.005 for the Sinkhorn's algorithm [15]. We set $\varepsilon_1$ as 1/3 and $\varepsilon_2$ as 1/2. We set $\tau$ as 1.0 in Eq. (9). We optimize the model parameters through Adam [30] with 200 epochs and learning rate as 0.002. We highlight that we use the above same hyperparameters for all reported experiments to demonstrate the stability of our FASTopic.

## E  Interpreting Learned Topic and Word Weights

As aforementioned, our FASTopic learns $\mathbf{s}$, the weight of each topic within the document collection and $\mathbf{u}$, the weight of each word within all topics. Figure 8 plots the relationship between the topic/word weights and the word frequency in the collection. Generally, topics with higher weights include top words of higher frequency, and words with higher weights also exhibit higher frequency. These observations confirm the validity of learned topic and word weights.

## F  Influence of Document Embedding Model

We investigate the influence of document embedding models. Apart from the mentioned `all-MiniLM-L6-v2`, we also experiment with `all-mpnet-base-v2` and `all-distilroberta-v1` as document embedding models in Tables 10 and 11. We notice that the performance is overall stable across these models. Moreover, the performance grows with `all-mpnet-base-v2` and `all-distilroberta-v1`, especially on document clustering. This is because these two produce document embeddings of relatively higher quality [54].

## G  Running Time Breakdown

To precisely compare BERTopic and FASTopic, we break down their running time. Table 12 shows that they both load document embeddings, but BERTopic takes more steps. BERTopic has to reduce embedding dimensionality, cluster embeddings, and compute word weights; in contrast, our FASTopic enjoys faster training. This is because FASTopic employs Sinkhorn's algorithm to solve the optimal transport, which is quite fast as proven by previous studies [15, 22]. Moreover, its objective is simple and straightforward as it only optimizes four parameters: topic and word embeddings and their weight vectors (Eq. (8).

Table 13: Topic quality results of coherence ($C_V$) and diversity (TD) under different dataset sizes ($N$) of WoS. The best is in **bold**.

| Model | N=15k CV | N=15k TD | N=20k CV | N=20k TD | N=25k CV | N=25k TD | N=30k CV | N=30k TD | N=35k CV | N=35k TD | N=40k CV | N=40k TD |
|---|---|---|---|---|---|---|---|---|---|---|---|---|
| LDA-Mallet | ‡0.354 | ‡0.862 | ‡0.353 | ‡0.850 | ‡0.352 | ‡0.861 | ‡0.357 | ‡0.858 | ‡0.354 | ‡0.860 | ‡0.360 | ‡0.849 |
| NMF | ‡0.386 | ‡0.413 | ‡0.385 | ‡0.439 | ‡0.393 | ‡0.444 | ‡0.396 | ‡0.444 | ‡0.389 | ‡0.442 | ‡0.394 | ‡0.446 |
| BERTopic | ‡0.438 | ‡0.639 | ‡0.441 | ‡0.646 | ‡0.438 | ‡0.631 | ‡0.438 | ‡0.633 | ‡0.436 | ‡0.639 | ‡0.432 | ‡0.625 |
| CombinedTM | ‡0.390 | ‡0.562 | ‡0.390 | ‡0.589 | ‡0.381 | ‡0.584 | ‡0.382 | ‡0.611 | ‡0.385 | ‡0.607 | ‡0.388 | ‡0.616 |
| GINopic | ‡0.366 | ‡0.607 | ‡0.367 | ‡0.630 | ‡0.362 | ‡0.660 | ‡0.362 | ‡0.690 | ‡0.364 | ‡0.678 | ‡0.364 | ‡0.719 |
| ProGBN | ‡0.375 | ‡0.842 | ‡0.370 | ‡0.856 | ‡0.378 | ‡0.881 | ‡0.378 | ‡0.866 | ‡0.372 | ‡0.867 | ‡0.373 | ‡0.872 |
| HyperMiner | ‡0.356 | ‡0.636 | ‡0.353 | ‡0.665 | ‡0.353 | ‡0.651 | ‡0.357 | ‡0.719 | ‡0.352 | ‡0.725 | ‡0.359 | ‡0.679 |
| ECRTM | 0.440 | **1.000** | ‡0.442 | **1.000** | ‡0.442 | **1.000** | ‡0.447 | **1.000** | ‡0.445 | **1.000** | 0.452 | **1.000** |
| FASTopic | **0.451** | 0.999 | **0.456** | **1.000** | **0.458** | **1.000** | **0.455** | **1.000** | **0.454** | **1.000** | **0.460** | **1.000** |

Table 14: Document clustering results of Purity and NMI under different dataset sizes ($N$) of WoS. The best is in **bold**.

| Model | N=15k Purity | N=15k NMI | N=20k Purity | N=20k NMI | N=25k Purity | N=25k NMI | N=30k Purity | N=30k NMI | N=35k Purity | N=35k NMI | N=40k Purity | N=40k NMI |
|---|---|---|---|---|---|---|---|---|---|---|---|---|
| LDA-Mallet | ‡0.656 | ‡0.331 | ‡0.637 | ‡0.323 | ‡0.638 | ‡0.319 | ‡0.638 | ‡0.318 | ‡0.639 | ‡0.317 | ‡0.627 | ‡0.314 |
| NMF | ‡0.550 | ‡0.228 | ‡0.531 | ‡0.205 | ‡0.533 | ‡0.209 | ‡0.531 | ‡0.209 | ‡0.533 | ‡0.204 | ‡0.524 | ‡0.196 |
| BERTopic | ‡0.642 | ‡0.322 | ‡0.645 | ‡0.316 | ‡0.635 | ‡0.314 | ‡0.641 | ‡0.310 | ‡0.641 | ‡0.308 | ‡0.638 | ‡0.304 |
| CombinedTM | ‡0.661 | ‡0.341 | ‡0.662 | ‡0.340 | ‡0.656 | ‡0.336 | ‡0.658 | ‡0.334 | ‡0.658 | ‡0.330 | 0.664 | ‡0.331 |
| GINopic | ‡0.634 | ‡0.301 | ‡0.630 | ‡0.294 | ‡0.633 | ‡0.294 | ‡0.631 | ‡0.289 | ‡0.637 | ‡0.290 | ‡0.632 | ‡0.284 |
| ProGBN | ‡0.611 | ‡0.326 | ‡0.618 | ‡0.328 | ‡0.615 | ‡0.318 | ‡0.623 | ‡0.322 | ‡0.610 | ‡0.313 | ‡0.608 | ‡0.317 |
| HyperMiner | ‡0.643 | ‡0.357 | ‡0.639 | ‡0.352 | ‡0.632 | ‡0.349 | ‡0.640 | ‡0.339 | ‡0.622 | ‡0.325 | ‡0.626 | ‡0.336 |
| ECRTM | ‡0.649 | ‡0.355 | ‡0.649 | ‡0.352 | ‡0.647 | ‡0.350 | ‡0.644 | 0.351 | ‡0.644 | ‡0.346 | ‡0.646 | 0.346 |
| FASTopic | **0.679** | **0.369** | **0.672** | **0.364** | **0.672** | **0.363** | **0.671** | **0.354** | **0.669** | **0.353** | **0.670** | **0.351** |

Table 15: Topic quality results of $C_V$ (topic coherence) and TD (topic diversity) under different vocabulary sizes ($V$) of WoS. The best is in **bold**.

| Model | V=20k CV | V=20k TD | V=30k CV | V=30k TD | V=40k CV | V=40k TD | V=50k CV | V=50k TD |
|---|---|---|---|---|---|---|---|---|
| LDA-Mallet | ‡0.351 | ‡0.855 | ‡0.350 | ‡0.843 | ‡0.354 | ‡0.833 | ‡0.356 | ‡0.849 |
| NMF | ‡0.397 | ‡0.431 | ‡0.394 | ‡0.415 | ‡0.392 | ‡0.439 | ‡0.396 | ‡0.449 |
| BERTopic | ‡0.463 | ‡0.676 | ‡0.443 | ‡0.675 | ‡0.384 | ‡0.680 | ‡0.400 | ‡0.672 |
| CombinedTM | ‡0.399 | ‡0.648 | ‡0.392 | ‡0.690 | ‡0.389 | ‡0.725 | ‡0.397 | ‡0.742 |
| GINopic | ‡0.371 | ‡0.623 | ‡0.370 | ‡0.718 | ‡0.368 | ‡0.713 | ‡0.376 | ‡0.725 |
| ProGBN | ‡0.370 | ‡0.767 | ‡0.370 | ‡0.766 | ‡0.386 | ‡0.634 | ‡0.381 | ‡0.614 |
| HyperMiner | ‡0.356 | ‡0.661 | ‡0.352 | ‡0.646 | ‡0.354 | ‡0.637 | ‡0.357 | ‡0.583 |
| ECRTM | 0.467 | ‡0.931 | ‡0.409 | ‡0.952 | ‡0.386 | ‡0.895 | ‡0.358 | ‡0.853 |
| FASTopic | **0.470** | **1.000** | **0.467** | **0.979** | **0.464** | **0.934** | **0.460** | **0.917** |

Table 16: Document clustering results of Purity and NMI under different vocabulary sizes ($V$) of WoS. The best is in **bold**.

| Model | V=20k Purity | V=20k NMI | V=30k Purity | V=30k NMI | V=40k Purity | V=40k NMI | V=50k Purity | V=50k NMI |
|---|---|---|---|---|---|---|---|---|
| LDA-Mallet | ‡0.637 | ‡0.332 | ‡0.640 | ‡0.334 | ‡0.636 | ‡0.326 | ‡0.644 | ‡0.331 |
| NMF | ‡0.548 | ‡0.229 | ‡0.545 | ‡0.233 | ‡0.560 | ‡0.240 | ‡0.552 | ‡0.235 |
| BERTopic | ‡0.660 | ‡0.338 | ‡0.655 | ‡0.333 | ‡0.644 | ‡0.332 | ‡0.652 | ‡0.331 |
| CombinedTM | ‡0.656 | ‡0.340 | ‡0.667 | ‡0.349 | ‡0.661 | ‡0.346 | ‡0.663 | ‡0.347 |
| GINopic | ‡0.629 | ‡0.302 | ‡0.624 | ‡0.303 | ‡0.634 | ‡0.303 | ‡0.636 | ‡0.304 |
| HyperMiner | ‡0.639 | 0.363 | ‡0.629 | ‡0.366 | ‡0.633 | ‡0.360 | ‡0.630 | ‡0.356 |
| ProGBN | ‡0.620 | ‡0.338 | ‡0.625 | ‡0.335 | ‡0.641 | ‡0.341 | ‡0.653 | ‡0.351 |
| ECRTM | ‡0.649 | ‡0.341 | ‡0.658 | ‡0.346 | ‡0.662 | ‡0.340 | ‡0.654 | ‡0.342 |
| FASTopic | **0.673** | **0.368** | **0.676** | **0.369** | **0.676** | **0.373** | **0.679** | **0.376** |

# H  Lists of Discovered Topics

Here we list all the discovered topics of different models. Compared to prior CombinedTM and BERTopic, FASTopic anchors topics with more specific and relevant words. Specifically in the NeurIPS dataset, the topics of CombinedTM and BERTopic usually contain high-frequency words like "algorithm", "data", "model", "learning", "neural", and "network". Admittedly they are related to some extent, but they are too general to anchor topic semantics [3, 4]. For instance, Topic#30 of BERTopic is about chip manufacturing, but contains general words like "neural", and "weight". In contrast, Topic#28 of FASTopic includes more specific ones like "silicon", "transistor", "cmos", and "neuromorphic". In the Wikitext-103 dataset, Topic#7 of BERTopic is about species, but has general words "known", "long", "large", "small", and "white". Differently Topic#2 FASTopic covers specific words "breeding", "prey", "subspecies", and "predators". See quantitative results of discovered topics in Sec. 4.2.

## CombinedTM (NeurIPS dataset)

#1: gradient dual convex convergence regularized accelerated proximal descent convexity smooth smoothness following generalized optimization mirror

#2: clustering laplacian matrix spectral eigenvectors clusters dimensional means matrices kernel graphs corresponding analysis reduction manifold

#3: behind contradicts proxy expectations negatives sam problematic exponentiated providing dumitru french equals addressing defines yuval

#4: network output parr input graduate units activation aki rumelhart activations mcclelland perturbation net devices backpropagation

#5: semantic text language set object example relations examples context caption sense target annotations word label

#6: policy value reinforcement state mdps optimal trajectory actions pomdp policies mdp iteration reward action horizon

#7: topic document modeling model latent lda collapsed chinese topics prior specific stick corpus distribution word

#8: image object shape friston images segmentation objects patches use videos bounding using features adversarial part

#9: admit negatives sit exponentiated completes addressing yuval dumitru proxy french upon virginia parallelize unfolding prints

#10: neurons alexander neuron plasticity cortical inhibitory rule mediated inputs firing patterns synapses synaptic input dendritic

#11: model attention lstm sequence models rnns rnn lstms modeling arxiv memory topic long sequential different

#12: loss complexity regret arm best ranking setting armed learning algorithms confidence bounds upper boosting bound

#13: exponentiated node variables given tree sampling pomdp probability factor variable finite gibbs new number state

#14: fire fig stimulus stable increasing head strength plasticity behind timing deg contrast trains spikes cells

#15: gaussian prior process likelihood latent posterior gamma covariance standard observations providing gas mean processes likelihoods

#16: clustering clusters number items random size cluster data algorithms acm algorithm users item means maximum

#17: drop measures empirical let positive learnability wrt theorem now generalization following boosting defined since uniform

#18: support adaboost space covering initiation inner olkopf margin leads vapnik mistakes notation risk conventions taylor

#19: attention temporal spatial hop two similar visual level morrison second video frame encouraged predict spatiotemporal

#20: units hubbard output net morgan stack rumelhart symbolic science internal weight weights epochs letters connectionist

#21: log complexity theorem case algorithm lemma pages regret bounds active upper let bound algorithms losses

#22: variables causal message node edges variable marginal marginals propagation inference chain sum degree graphical nodes

#23: classification distance positive metric framework label multi codes task learning semi tasks accuracy kernel test

#24: pointed promoting problematic stack home place accumulated unknowns fibers graepel thresholded locomotion aliasing hamilton tseng

#25: patch color scenes prototypes shape object patches scene rst voxels tracking image objects pixels audio

#26: regret algorithms strategies games mdps player armed subsequently algorithm confidence equilibrium best online known mdp

#27: unfolding output morgan inductively jacobs biases thesis realizes smoother iearning problematic forces species protein shortcoming

#28: observer sources speech source audio snr carbonell gardner signals pitch unpublished location middle perceptual harmonic

#29: learning generative training use output samples rostamizadeh objective adversarial deep arxiv dropout batch networks work

#30: qin problematic admit expectations pooled providing addressing yuval french analysed drop carlos crucially regressors collectively

#31: set node clustering tree sets clusters means search algorithms graph points given cost number solution

#32: stimulus fig coding noise responses correlation natural sensory estimated decoding information population agency neurons tricky

#33: functions upper theorem bounds case map define polynomial influence show lemma set following max defined

#34: motor left field movements location motion perceptual bottom right turned locations eeg bci humans bar

#35: samples log posterior variational gaussian likelihood latent bayesian approximate test elbo prior inducing data monte

#36: norm lasso solution problem matrix following matrices low sparse constraint rank sparsity methods global selection

#37: providing proxy addressing admit sam problematic qin contradicts carlos virginia french webpage exponentiated analysed dumitru

#38: sample statistical estimation distributions statistics estimator lasso dimensional log copula condition estimators theorem families estimating

#39: intelligent morgan expectations complete unification host damping interfacing required thesis ingredients sam drop triple agency

#40: master transfer results dataset image images classification imagenet plain representation ranging deep generator convolutions based

#41: making behavior plasticity decision observer estimated change sensory time effect trials term trains fit trial

#42: behind guide providing appended problematic proxy addressing admit qin negatives yuval orabona french carlos intelligent

#43: network size networks pooling layer convolution layers accuracy weights without gradient sequence deep batch neural

#44: state agents agent actions expert reinforcement qin task goal decisions based rewards skills world observation

#45: matrix power tensor columns completion rank matrices sparsity via decomposition singular iteration high iterations analysis

#46: mediated guide damping vivo intelligent problematic addressing cerebellum axon crucially mostafa behind dumitru unfolding pull

#47: guide rbf database distance classifier sets classifiers roc faces classification validation vectors performed euclidean cascade

#48: loss norm statistical convex following theorem convexity functions excess term conditions ferrari mirror observes condition

#49: kxt cells inhibitory selectivity cortical qin neurosci cones cell bar effects christoph address sam bottom

#50: machine algorithms problem pull functions objective learning loss set optimization solution svm constraints boosting methods

## BERTopic (NeurIPS dataset)

#1: learning model data set algorithm using function one training network two figure time number models

#2: kernel learning active error algorithm kernels svm training examples set function theorem data bound margin

#3: convex convergence gradient optimization algorithm stochastic descent algorithms loss regret online problems problem rate method

#4: spike neurons neuron synaptic firing spikes neural time model spiking activity population stimulus input information

#5: policy state reward agent action learning reinforcement value function game agents actions games optimal algorithm

#6: speech recognition speaker training neural network recurrent sequence input model output word rnn networks hidden

#7: visual motion eye cells saliency orientation model spatial image receptive neurons figure response cell stimulus

#8: sparse lasso pca matrix norm algorithm recovery problem data principal sparsity log rank analysis solution

#9: graph graphical variables inference map graphs node algorithm models tree nodes edge set problem log

#10: gaussian posterior model process data bayesian models state time covariance likelihood prior distribution function mean

#11: graph manifold metric distance points data embedding nearest learning graphs neighbor space dimensional kernel local

#12: network networks neural units input weights hidden output function learning layer weight one rules unit

#13: deep layer networks training convolutional layers network gan image learning neural arxiv images generative cnn

#14: topic word words topics lda document model documents models language dirichlet latent data distribution corpus

#15: model memory decision stimulus reward response trial figure stimuli learning task trials models two time

#16: brain eeg subject fmri subjects functional data connectivity voxels bci spatial voxel time activity motor

#17: image images object features visual model shot classes question training class feature learning set objects

#18: regret arm bandit arms bandits ucb algorithm bound reward armed problem log action setting exploration

#19: matrix rank tensor norm completion low entries matrices algorithm decomposition problem factorization tensors theorem recovery

#20: policy function state optimization value belief pomdp optimal uncertainty algorithm pomdps search mdp reward problem

#21: variational mixture inference posterior log models data dirichlet distribution gradient bayesian parameters likelihood model components

#22: clustering clusters cluster spectral means algorithm cut data points graph matrix problem partition set number

#23: video motion pose frames frame model tracking image temporal human body object using flow figure

#24: task tasks learning domain target multi transfer source data adaptation training domains feature multitask problem

#25: ranking user item rank items users query ratings model pairwise collaborative top algorithm rankings matrix

#26: label labels unlabeled supervised labeled classification data learning semi class loss set examples multiclass problem

#27: control controller motor trajectory model arm movement forward system learning network robot time inverse movements

#28: auditory frequency sound sounds cochlear signal neurons model time stimuli localization responses stimulus system response

#29: nodes influence network node social networks community communities time model edges graph link edge algorithm

#30: chip circuit analog voltage neuron vlsi synapse input circuits current gate digital output weight neural

#31: image images resolution denoising reflectance depth blur pixel noise color pixels scene shading deconvolution scale

#32: density kernel test sample estimation mmd tests estimator statistic characteristic distribution null kernels two samples

#33: patient survival patients disease model time clinical risk data models cancer decision process event individual

#34: causal variables graph data model treatment observational interventions models discovery discrimination causes directed set effects

#35: ica separation source sources signals blind signal independent mixing matrix components algorithm component speech mixtures

#36: protein gene genes proteins expression prediction sequence sequences model data species amino structure features binding

#37: sampling hmc monte carlo hamiltonian sample distribution mcmc samples chain scan smc proposal stochastic dynamics

#38: privacy private differentially differential mechanism data algorithm log theorem party utility let protocol distribution queries

#39: sparse coding basis sparsity coefficients image prior dictionary wavelet overcomplete model reconstruction data slab images

#40: face facial images faces recognition image human features cnn feature pca verification subjects classification svm

#41: submodular functions function algorithm greedy set approximation maximization monotone submodularity solution algorithms problem streaming representatives

#42: price market revenue prices auctions regret bid reserve strategic profit maker markets algorithm optimal trading

#43: workers worker crowdsourcing labels voting tasks task label annotators alternatives crowd majority true items model

#44: music musical harmonic notes note rules rule pitch song beat tags time system structure signal

#45: hashing hash codes binary bits lsh hamming bit similarity precision search data functions retrieval query

#46: choice preferences utility preference model user decision rankings options users set choices models item option

#47: memory capacity associative memories stored patterns hopfield network pattern recall storage networks sdm number retrieval

#48: trees tree decision forests split leaf node forest training dags random nodes breiman feature splits

#49: routing traffic call arrival channel load rate time policy service mobile calls state control loss

#50: relational entities relations link entity relation links model tensor factorization relationships models data learning rank

## FASTopic (NeurIPS dataset)

#1: wahba steinwart sriperumbudur sollich rasch sobolev christmann integrable heteroscedastic blanchard fukumizu unregularized qui mcdiarmid functionals

#2: graphs vertex vertices edges graph trees tree edge node nodes message directed loopy lifted treewidth

#3: ranking users user items item documents topics topic lda document social web rankings crowdsourcing votes

#4: robot controller controllers iearning atkeson backup bradtke satinder planner kaelbling doina bellemare antonoglou aamas pendulum

#5: deep convolutional layers cnn layer rnn encoder lstm trained recurrent dropout architecture arxiv architectures bengio

#6: crammer halfspaces perceptron multilabel koby disagreement beygelzimer classi holdout mistake gentile aggressive littlestone queried claudio

#7: net units network activation networks unit nets capacity back hidden connection multilayer patterns backpropagation output

#8: dauphin desjardins ganguli wojciech gulcehre razvan rgen glorot barham theano pascanu abadi dahl arjovsky zaremba

#9: cognitive automaton verbal pollack teach hillsdale recalled psychological cards cognition psychology infants teaching chess episodic

#10: carlo monte hyperparameters salimans hmc titsias ranganath gans elbo ais langevin vae radford hamiltonian rezende

#11: waibel frasconi widrow squashing holmdel giles jaitly zipser retraining bahdanau freeze microstructure retrained ronan denver

#12: trajectory dynamics trajectories control dynamical dynamic state states transition transitions sequences system kalman temporal forward

#13: price regret adversary round market revenue prices auctions hedge bid markets profit ads multiarmed reserve

#14: receptive lgn striate geniculate retinotopic topography movshon afferents eero ventral ocular parietal orientation hubel lond

#15: stimulus stimuli cortex cells neurons activity sensory brain responses neuron cell response cortical neuroscience trial

#16: classifier classifiers svm unlabeled margin boosting labeled classification label labels supervised classes class examples svms

#17: manifold kernels kernel dimensionality eigenvectors laplacian eigenvalues metric euclidean embedding principal pca tangent isomap eigenfunctions

#18: mika smo kandola holloway scholkopf fukunaga quinlan varma misclassifications canu tsang grandvalet wrapper zien mlrepositoryhtml

#19: causal model structure models group across data individual modeling features specific different three used experiment

#20: amino acid acids conserved jojic szeliski mol registration isometric proteins analyzers tomasi molecular shading tissue

#21: clustering clusters cluster hashing hash lsh linkage anomaly clusterings dissimilarities agglomerative kmeans indyk luxburg charikar

#22: estimators estimator regression multivariate covariance density estimating variance additive gaussian parametric asymptotic estimation bias estimates

#23: recovery tensor rank sparsity entries sparse completion lasso singular columns matrices norm matrix factorization decomposition

#24: scene object image images pixels pixel segmentation pose vision face cvpr patches color objects video

#25: haar printed strokes eigenfaces karayev photographs downsampling zip satheesh shelhamer resized sermanet maire caffe bruna

#26: ancestral propositional kemp dumais wallach grammars predicate jurafsky syntax darwiche noun predicates grammar parser naacl

#27: reaction death events triggering survival diffusion viral mice reactions longitudinal regulatory durations progression species event

#28: chip circuit analog voltage circuits vlsi silicon cmos transistor transistors chips voltages neuromorphic axon fabricated

#29: winther buntine opper paisley andriy digamma moitra niranjan andrieu knowles cumulant kappen doucet barber minka

#30: alp boyan ghavamzadeh puterman nord dimitri ortner lille lazaric optimism csaba szepesvari tsitsiklis restart polyak

#31: bounds minimax sup risk inequality privacy corollary lemma bounded bound theorem proof inf holds upper

#32: hebert ramanan urtasun articulated schiele volumetric triggs lampert occlusions indoor hoiem animation mathieu saenko metz

#33: posterior gibbs inference bayesian priors mixture dirichlet likelihood latent variational sampler mcmc marginal probabilistic poisson

#34: policy reward agent actions policies reinforcement agents mdp games player action exploration planning rewards arm

#35: nongaussian diagnostics candela snelson ard visualisation warped imputation reversible duvenaud vague aic dbns carlin neil

#36: bifurcation chaos basins chaotic attractors oscillates oscillate basin attraction landscapes interconnections interconnection sompolinsky tank salesman

#37: speech speaker hmm audio acoustic phoneme speakers phone phonetic utterances utterance spoken female voice recognizer

#38: spikes spiking spike synaptic firing synapses membrane trains postsynaptic connectivity stdp calcium hippocampal epsps potentiation

#39: language word semantic words text sentence embeddings category captions phrase mikolov caption captioning answering sentences

#40: eye motion velocity saliency gaze fixation saccades saccadic optic itti observers photoreceptor salience distractors photoreceptors

#41: caruana women expertise disorders atlas diagnosis diagnostic pet mitchell prototype classifications discriminating niculescu clues exercise

#42: rip donoho dantzig isometry candes incoherent obozinski johnstone emmanuel baraniuk negahban fazel parrilo caramanis vershynin

#43: darken soviet fitness beating santosh lms morrison purdue schraudolph submission splines penrose sherman placements hassibi

#44: descent proximal sgd primal dual coordinate strongly convex convergence smooth gradient accelerated svrg kxt nesterov

#45: brownian elliptical bivariate dunson breakdown climate covariates forecasting econometric forecast semiparametric econometrics forecasts observational cdf

#46: submodular approximation solution algorithms algorithm problem functions optimization function problems approximate objective constraints linear greedy

#47: learner active queries online decision hypothesis query rule strategy mechanism adaptive cost every concept making

#48: sketching threads mahoney drineas woodruff parallelizing parallelize preconditioning cpus cores thread tak speedups parallelization richt

#49: auditory eeg sources ica signals sounds bci meg cochlear khz hearing microphone ear acoust acoustical

#50: naor mossel lov iyer combinatorics submodularity mcsherry asz karp bilmes talwar feige nemhauser karger wolsey

## CombinedTM (Wikitext-103 dataset)

#1: general war confederate washington fort massachusetts york grant army kentucky william convention men states united

#2: storm september october upgraded southwest day northeastward strengthened convection briefly tropical bermuda island southeast dissipating

#3: second lap place drivers driver third ahead fourth position edwards time stage stops classification caution

#4: also music released one film disney million animation time show made year first video new

#5: ride station closed location ownership historic train rail restaurant parking services building norwegian stations underground

#6: video song top number chart music hot dancers performance carey madonna week dancing performed love

#7: city island unk war river area population many also region san coast spanish settlements century

#8: queen prince king made years royal charles england london william death later family henry father

#9: title match team championship world won tag defeated ring wrestling face joe cage referee raw

#10: court courts right amendment criminal clause requires cent public person singapore law case without dollar

#11: stone built century house period william dates wall chapel tower henry buildings gothic england site

#12: species years populations found cause large known prey conservation water feed muscle individuals many hunting

#13: first time one made year world years three new two team won also season second

#14: heads residential ends junction southeast redesignated county designated portion splits route briefly woods divided interchange

#15: constantinople chronicle hungary byzantine prince sources empire count emperor conflict kingdom conquest rebellion brother commanded

#16: brigade army division unit forces general infantry battalion casualties hill men artillery corps flank commanded

#17: york year time said family gay television years new life later american show award obama

#18: game player games gameplay playstation version guitar soundtrack hero features mode original multiplayer ign mario

#19: stroke boat rowing olympic cox progressed ahead oxford excluding finishing toss referred athlete push gold

#20: listing musician tragic albums drums personnel notes vocal liner garde drummer instrumental duo sounds soundtrack

#21: life jesus work god women wrote philosophy moral spiritual ideas scientific book argues religious views

#22: party national members leadership leader state government conservatives college assembly elections minister liberal support prime

#23: season pitched games played year baseball basketball team league giants ncaa home signed record named

#24: education degree board professor enrolled mayor serve attended deputy worked founder associate massachusetts district assembly

#25: unk known many also used dynasty period speakers horses century ancient greek word found form

#26: episode nielsen scully tom andy broadcast jenna tracy reviews scene files simpsons creator watched funny

#27: battle two british three fire fleet men made day line ships island left warships position

#28: music harrison band bands songs composers work musician musicians album progressive beatles musical folk recorded

#29: storm flooded damage island precipitation surge downed tornado water homes damaged along tropical near coast

#30: story characters character anime quest game manga final original protagonists novels voiced player volumes protagonist

#31: tons armor steam knots adriatic battery armored turrets displaced laid aft monitor boilers consisted stern

#32: album band albums record songs released track copies studio group chart live release recording records

#33: discovery ignore commemorating nomenclature outlined indirectly planet divide addressing garde earnest discovered masses shift judged

#34: air pilots aircraft wing squadron aviation flying fighter training service flew nuclear squadrons flight pilot

#35: club made arsenal played minute striker substitute half united liverpool side scored england cricket first

#36: city stadium club college located company students sports football school include campus hosted largest schools

#37: soviet moscow war nazi german government finnish line polish regime germany country jews electric hitler

#38: film movie bond role reviews grossed scenes films critics production grossing india rotten picture director

#39: game bowl first yards time quarter also kick season line offense possession tech alabama virginia

#40: mushroom bearing shape gill growing flesh attached color taste orange diocese fungus epithet cap fine

#41: line river along bridge railway miles freeway junction county part rail road interchange built mile

#42: hot chart lewis charts tempo digital latin vocal dancing forty singles billboard debuted background charted

#43: nielsen bart adrian simpson arrives broadcast reference peter burns wallace watching olds fox realizes pearson

#44: stable properties known used material hydrogen curve protein experiment form applications processes process formula space

#45: fleet turrets british seas ships port tons ship baltic class convoy armor fire aft lasted

#46: ignore judged commemorating monopoly divide garde outlined enhanced addressing constantly obituary unprecedented earnest exploit busy

#47: editions magazine artist book critic photographs published art paint publisher stories publishers notes publication read

#48: species conservation brown populations mature blue feed slightly typical tree legs individuals tail iucn eggs

#49: relationship ben neighbours episode soap storylines character amy rachel mitchell davies finale episodes get producers

#50: flooded preparations downed convection upgraded warning downgraded homeless affected storm eye meteorological originated decreased ashore

## BERTopic (Wikitext-103 dataset)

#1: unk also one first new two time film song later game war three album city

#2: album song band music songs chart number released video single rock tour recording track madonna

#3: episode show series season episodes homer character television viewers simpsons scully said scene bart doctor

#4: ship ships squadron war fleet aircraft guns air navy british japanese force two battle gun

#5: season team game games first league runs innings played career second scored test won record

#6: highway route road state county north freeway creek east interchange street intersection along avenue south

#7: species unk genus found birds known shark females males long large brown small white common

#8: storm tropical hurricane winds cyclone mph damage depression rainfall wind typhoon system landfall september flooding

#9: film films bond role character unk best production disney actor story also movie million director

#10: division army infantry german brigade battle forces troops battalion british corps north war attack regiment

#11: game player games players gameplay fantasy released characters character nintendo final version series development release

#12: king scotland henry england edward century scottish royal unk william english island son queen bishop

#13: castle bridge town century built house river building canal area local south road centre west

#14: art book stories painting novel work unk comics published poem fiction works story issue magazine

#15: election governor party state president campaign government republican senate kentucky democratic new house political elected

#16: race lap car stage cambridge oxford racing drivers team lead second won races riders points

#17: unk anime manga series also one language released greek character first english century story used

#18: polish unk soviet poland croatia russian war army emperor byzantine military empire forces hungary government

#19: unk temple government city chinese singapore dynasty china state emperor also court india arab minister

#20: club league cup season football goal match scored team arsenal goals stadium first final win

#21: station railway line trains train ride london services passenger roller class service stations opened railways

#22: music opera orchestra composer musical works symphony unk piano work gilbert sullivan first theatre performance

#23: nuclear unk atomic laboratory compounds used element metal project physics hydrogen chemical energy research university

#24: star planet earth sun planets stars orbit jupiter solar mass magnitude surface system dwarf moon

#25: match championship wrestling tag event team raw defeated ring title champion world angle triple feud

#26: church god unk century congregation christian churches pope religious catholic one christ also building moral

#27: airport line norwegian station norway unk party trains tunnel swedish service services started built meters

#28: disease protein cells unk cell symptoms risk blood dna treatment acid cause virus cases infection

#29: river park dam volcanic water creek area lake national flows unk feet valley canyon mountain

#30: school students university college campus student education georgia schools research program academic faculty tech building

#31: court chicago city state park indiana states county supreme building district united river illinois federal

#32: company restaurant chicken food king unk product beer products wine menu new restaurants chain ingredients

#33: spanish texas mexican san city government unk spain houston mexico bay juan plaza puerto political

#34: trek enterprise episode star space crew apollo series mission spacecraft kirk nasa first season earth

#35: coins flag coin dollar silver design struck cent gold pieces eagle dollars statue reverse flags

#36: oil darwin bank evolution natural species unk plants billion energy investment organisms evolutionary selection animals

#37: horses breed horse breeds sheep breeding dogs dog bred used registered century riding white unk

#38: hotel building library theatre mall center square feet floor new art museum room theater city

#39: police said murder case evidence fire murders trial found death court told prison government people

#40: apple windows software system data console unk playstation microsoft nintendo user users hardware device released

#41: formula function theory space numbers number matrix frequency unk example used mechanical constant mathematical linear

#42: flight airline airlines aircraft air boeing airport crash flights passengers accident crew international aviation plane

#43: radio network stations television station broadcasting digital programming mutual span broadcast paramount news channel cable

#44: pedro brazil brazilian rio emperor government argentina naval navy war ships chile portuguese portugal cabinet

#45: resident evil god game war playstation leon umbrella released iii player series character claire games

#46: adriatic croatia traffic toll route port section interchange river areas sea kilometres rest construction basin

#47: football women team fifa national cup peru country world tournament players association teams competition played

#48: spider man peter parker film amazing character sony comic harry webb jane comics mary miles

#49: children show street television producers workshop educational research curriculum viewers production lesser clues goals productions

#50: harry potter book film books ron series million magic children philosopher novel phoenix stone released

## FASTopic (Wikitext-103 dataset)

#1: kentucky virginia massachusetts ohio indiana confederate lincoln illinois pennsylvania congressman sherman missouri jefferson democrat congressional

#2: species birds males breeding females animals bird fish breed prey eggs breeds nest subspecies predators

#3: often movement many life even unk social among others age popular children years become considered

#4: railway creek bridge lake river park road county valley street miles construction opened canal line

#5: ships ship fleet guns admiral tons torpedo navy hms inch naval gun deck cruiser knots

#6: students university college school campus chicago student education program research schools business company arts building

#7: viewers jenna viewership storylines storyline dwight ratings soap tracy jim nbc fringe alec timeslot finale

#8: homer simpsons bart scully lisa files fox dana households springfield burns leslie nielsen aired manners

#9: goddess deity dialect ingredients chicken wine folklore sheep hindu cooking gods rituals silk pig bread

#10: vampire villains antagonist creature kills villain escapes backstory jake demons monsters kill demon stan flees

#11: century church temple population centre india chinese roman scotland ancient period site region built local

#12: tower floor walls storey courtyard architects roof architectural constituency brick excavations architect carved castle manor

#13: baseball basketball pitcher rookie freshman pitching overtime ncaa espn assists tigers leagues hockey sophomore traded

#14: wrestling tag liverpool match raw ring goalkeeper striker ham referee footballer matches champion championship pinned

#15: planet planets volcanic jupiter magnitude orbital geological orbit minerals geology observatory cluster melting plateau dioxide

#16: cricket innings test olympic australia matches match bowling won race stage runs event win competition

#17: flotilla adriatic dockyard casemates amidships masts gunners austro keel sms conning broadside bombarded towed cruising

#18: mario computer software gamer eurogamer graphical puzzles informer consoles microsoft user arcade puzzle sonic interactive

#19: show comedy guest relationship audience interview really shows girl broadcast friends television think commented sex

#20: torrential thunderstorm inundated intensify intensifying outflow outages downgraded periphery saffir thunderstorms shelters disorganized intensification currents

#21: grossing screenplay theaters grossed filmmakers ebert screenwriter cinematographer tomatoes picture cinematography cinema paramount movies rotten

#22: trains train airport passengers passenger stations cars airline ride roller freight destinations railways airlines runway

#23: drummer bassist vocalist guitars vinyl riff nme riffs guitarist demos headlining unreleased keyboards labels dylan

#24:  clergy bishops ecclesiastical protestant cardinal theological priests monks diocese catholics theology teachings papal manuscripts catholicism

#25: narrator feminist novelist autobiographical reprinted prose poetic poets essays illustrations imagination anthology reader essay realism

#26: freeway interchange intersection highway terminus avenue passes crosses continues lane turns alignment heads interstate intersects

#27: mathematics curriculum economics professor thesis undergraduate lectures psychology lecture mathematical ethics nobel harvard physics journalism

#28: aircraft flight air flying fighter wing mission squadron pilot operations landing bomb bomber pilots raf

#29: disease risk treatment blood cases symptoms cell cells protein diagnosis clinical infection patients medical brain

#30: confluence flanking flourished strategically advocating headquartered undermine formulated overrun strife annexed affiliation landowners aristocracy bordered

#31: cadet howe sergeant scout scouts citation volunteered badge decorations bravery discharged instructor scouting trenches rifle

#32: cylinder engine specifications prototype machine capability weight mechanical barrel variants configuration manufactured manufacture muzzle wheel

#33: polish flag soviet poland russia russian croatia nationalist dutch republic jews countries serbian israeli socialist

#34: emperor reign king empire roman byzantine monarchy throne castles archbishop castle ruler monarch revolt persian

#35: film films episode cast character scenes script actor scene movie series episodes filming production director

#36: league football goals goal cup coach yard stadium yards scored club season teams players team

#37: overthrow communists coup bin partisan marched embassy pact peaceful factions faction exiled massacre negotiate rebel

#38: mary married sir william london wife thomas queen henry edward lord george died elizabeth charles

#39: lap cambridge oxford riders races rowing seconds cycling olympics rider race athletes drivers caution lengths

#40: infantry battalion brigade regiment troops army corps artillery forces soldiers division battle command attack wounded

#41: novel books stories book author works fiction art poem opera literary text poems published poetry

#42: album chart band song songs billboard albums guitar recording lyrics pop vocals singles madonna music

#43: tropical hurricane cyclone storm winds rainfall depression flooding intensity mph landfall wind damage typhoon utc

#44: energy earth mass formula surface carbon gas type hydrogen chemical data temperature systems example process

#45: prison murder police prosecution trial jury investigation murders guilty convicted alberta conviction sentence crime testified

#46: piano orchestra composer dancers conductor violin pianist symphony singers tenor orchestral composers composition duet concert

#47: gameplay fantasy nintendo playstation game player anime manga soundtrack xbox characters games players dragon mode

#48: lyrically certifications airplay pitchfork synth synthesizers remix remixes remixed catchy rapper vibe listings downloads slant

#49: cap fruit fungus spores phylogenetic taxonomic morphological clade hairs basal mushroom microscopic stem morphology epithet

#50: election party law government political court president minister senate republican democratic constitution rights committee elected

