# OpenReview forum: "FASTopic: Pretrained Transformer is a Fast, Adaptive, Stable, and Transferable Topic Model"
_NeurIPS.cc/2024/Conference — NeurIPS 2024 poster_

### Official Review · Reviewer_YWcC · 2024-06-27

**Soundness:** 3
**Presentation:** 3
**Contribution:** 3
**Rating:** 7
**Confidence:** 4

**Summary:**

The paper proposes FASTopic, a topic model using pre-trained document embeddings and embedding transport between documents and topics as well as words and topics. The minimized objective function is then a combination of the DSR and ETP which is optimized via finding topic, and word embeddings.

**Strengths:**

- Interesting idea which is well documented
- Initial experimental results show potential
- potential benefit of getting interpretable word-embeddings while training topic model

**Weaknesses:**

- Overall, I quite like the paper and the idea. However, I have some reservations, primarily due to the aggressive tone of the presentation, starting from the abstract. The paper repeatedly criticizes comparable methods, while presenting its own method as the ultimate and superior topic model. Such claims need robust support through experimental evidence. Although the experiments provide some validation, the tone of the paper creates a discrepancy, making the experimental results appear insufficiently convincing.

  Overall, more results are needed to back up the strong claims made in the paper.

  Below are some suggestions that, in my opinion, would support the claims more effectively. If these suggestions are unjustified, I would appreciate an explanation.


  **Minor:** I think the high-resolution plots are causing some trouble. I cannot really scroll through the paper smoothly.


1. **Include further models** -> ETM [1], ProdLDA [2], CEDC [3], CTMneg [4]. While I wouldn't suspect ETM/ProdLDA to be better than any other of the models, the simple models CTMneg and CEDC have shown to outperform BERTopic.

2. **Incorporate other evaluation metrics**: since evaluation is very difficult [5]. E.g. use some presented in [3] or [6].

3. **How do you measure training time for all of the models?**
   - 3.1) Are all steps, including the document encodings, part of the taken training time?
   - 3.2) Do you use the same encoding model for all comparison models (where applicable)?
   - 3.3) Given that FASTopic outperforms BERTopic in terms of speed, I wonder for how many epochs you are training your model, and what it is that takes so long in BERTopic? The document encoding steps are the same for both models and while dimensionality reduction takes some time using UMAP, I have a hard time believing that it is slower than training FASTopic for a reasonable amount of epochs.

4. **How did you choose the number of topics for BERTopic?** Did you use Kmeans instead of HDBSCAN or hierarchicaly reduced the number of topics?

5. **How many parameters does FASTopic have compared to the other neural models?**


[1] Dieng, A. B., Ruiz, F. J., & Blei, D. M. (2020). Topic modeling in embedding spaces. Transactions of the Association for Computational Linguistics, 8, 439-453.

[2] Srivastava, A., & Sutton, C. (2017). Autoencoding variational inference for topic models. arXiv preprint arXiv:1703.01488.

[3] Adhya, S., Lahiri, A., Sanyal, D. K., & Das, P. P. (2023). Improving contextualized topic models with negative sampling. arXiv preprint arXiv:2303.14951.

[4] Thielmann, A., Reuter, A., Seifert, Q., Bergherr, E., & Säfken, B. (2024). Topics in the haystack: Enhancing topic quality through corpus expansion. Computational Linguistics, 1-37.

[5] Hoyle, A., Goel, P., Hian-Cheong, A., Peskov, D., Boyd-Graber, J., & Resnik, P. (2021). Is automated topic model evaluation broken? the incoherence of coherence. Advances in neural information processing systems, 34, 2018-2033.

[6] Stammbach, D., Zouhar, V., Hoyle, A., Sachan, M., & Ash, E. (2023). Revisiting automated topic model evaluation with large language models. arXiv preprint arXiv:2305.12152.

**Questions:**

- Are the created word embeddings semantically meaningful? Would clustering the word embeddings e.g. [7] give meaningfull topics? If yes, this is an additional benefit which I would suggest to include in the paper.



[7] Sia, S., Dalmia, A., & Mielke, S. J. (2020). Tired of topic models? clusters of pretrained word embeddings make for fast and good topics too!. arXiv preprint arXiv:2004.14914.

**Limitations:**

Not needed

---

> ### Author Rebuttal · Authors · 2024-08-07
>
> Thank you for your insightful reviews! We appreciate you find our work interesting and results potential.
> We hope our responses can address your concerns and improve your rating.
>
> __Q1: comparison to more models__
>
> We emphasize __we have included the newest baselines__: HyperMiner(NeurIPS2022), ProGBN(ICML2023), ECRTM(ICML2023), GINopic(NAACL2024).
> Here we report the comparisons to CTMneg and CEDC:
>
> |Model|20NG||NYT||WoS||NeurIPS||ACL||Wikitext-103||
> |:-|:-|:-|:-|:-|:-|:-|:-|:-|:-|:-|:-|:-|
> ||$C_V$|TD|$C_V$|TD|$C_V$|TD|$C_V$|TD|$C_V$|TD|$C_V$|TD|
> |CTMneg|0.378|0.615|0.381|0.633|0.377|0.571|0.402|0.526|0.393|0.388|0.389|0.708|
> |CEDC|0.413|0.375|0.382|0.456|0.443|0.566|0.343|0.212|0.306|0.267|0.400|0.471|
> |__FASTopic__|__0.426__|__0.983__|__0.437__|__0.999__|__0.457__|__1.000__|__0.422__|__0.998__|__0.420__|__0.998__|__0.439__|__0.992__|
>
>
> We see although CTMneg is better than CombinedTM, FASTopic still outperforms them.
> Thank you for mentioning these important work. We have updated the paper and cited them.
>
> __Q2: include more evaluation metrics__
>
> Here we additionally report the results of Word Embedding Coherence (WEC) and Inversed Rank-Biased Overlap (IRBO) from Adhya_2023.
> WEC measures coherence with pretrained word embeddings, and IRBO measures diversity.
>
> |Model|20NG| |NYT| |WoS| |NeurIPS| |ACL| |Wikitext-103| |
> |:-|:-|:-|:-|:-|:-|:-|:-|:-|:-|:-|:-|:-|
> ||WEC|IRBO|WEC|IRBO|WEC|IRBO|WEC|IRBO|WEC|IRBO|WEC|IRBO|
> |LDA-Mallet|0.034|0.993|0.054|0.996|0.070|0.997|0.041|0.989|0.052|0.984|0.068|0.990|
> |NMF|0.040|0.987|0.028|0.978|0.081|0.983|0.045|0.971|0.045|0.970|0.069|0.976|
> |BERTopic|0.043|0.990|0.056|0.992|0.109|0.990|0.049|0.975|0.058|0.978|0.077|0.984|
> |CombinedTM|0.038|0.986|0.029|0.993|0.084|0.990|0.036|0.986|0.036|0.972|0.050|0.992|
> |GINopic|0.038|0.996|0.049|0.991|0.076|0.990|0.047|0.984|0.037|0.987|0.053|0.991|
> |ProGBN|0.044|0.967|0.052|0.961|0.103|0.985|0.042|0.886|0.054|0.964|0.070|0.940|
> |HyperMiner|0.038|0.988|0.052|0.990|0.077|0.990|0.047|0.997|0.045|0.997|0.065|0.988|
> |ECRTM|0.042|__1.000__|0.040|0.998|0.066|__1.000__|0.051|__1.000__|0.054|__1.000__|0.088|__1.000__|
> |__FASTopic__|__0.054__|__1.000__|__0.066__|__1.000__|__0.130__|__1.000__|__0.053__|__1.000__|__0.060__|__1.000__|__0.093__|__1.000__|
>
> The above shows our model also performs better on these metrics.
> Thank you for the suggestion. We have added these results to the paper.
>
> __Q3: the number of topics for BERTopic__
>
> We explain __we set the number of topics of BERTopic to be the same as the other models for fair comparisons.__
> Each model produces 50 topics in Table_1. Otherwise, it'll be unfair if BERTopic reduces the number of topics to 10.
>
> __Q4: how is the training time measured?__
>
> __We measure the training time from when the dataset is loaded until the training is finished.__
> The training time includes loading document embeddings.
>
> We clarify __we use the same document embeddings for each model when applicable for fair comparisons.__
> As mentioned in Section_D Line_574, we use `all-MiniLM-L6-v2` for document embeddings by default.
>
> __Q5: why FASTopic is faster than BERTopic__
>
> As mentioned in Appendix_D Line_578, we train FASTopic with 200 epochs.
> We break down the training time (seconds) as
>
> |BERTopic||
> |:-|:-|
> |Step 1: Load doc embeddings|7.10|
> |Step 2: Reduce dimensionality|23.13|
> |Step 3: Cluster doc embeddings|0.21|
> |Step 4: Compute word weights|1.97|
> |__Sum__|32.41|
>
> |FASTopic||
> |:-|:-|
> |Step 1: Load doc embeddings|7.10|
> |Step 2: Training|5.85|
> |__Sum__|12.95|
>
> We see BERTopic has to reduce embedding dimensionality, cluster embeddings, and compute word weights.
> In contrast, our model enjoys faster training.
> This is because it employs Sinkhorn's algorithm to solve optimal transport, which is quite fast as proven by previous studies (Cuturi_2013, Genevay_2019).
> Moreover, __its objective is simple and straightforward, optimizing only four parameters__: topic and word embeddings and their weight vectors (Eq(8)). This avoids the complicated encoders and decoders of VAE-based methods.
> Previous ECRTM also uses optimal transport, but its objective based on the complicated encoder and decoder slows it down.
>
> Thank you for the question. We've added more explanations to the paper.
>
> __Q6: the number of hyperparameters in FASTopic__
>
> We summarize the hyperparameters as follows:
>
> 1. __FASTopic__ has hyperparameters for Sinkhorn's algorithm $\varepsilon_{1}$ and $\varepsilon_{2}$ in Eq(5,6) and the temperature in Eq(9).
> 2. VAE-based models (CombinedTM, ProGBN, HyperMiner, ECRTM, GINopic) need more hyperparameters to set their encoders, decoders (dimensions, number of layers, and dropout), and prior distributions (Gaussian or Dirichlet).
> Some need more hyperparameters, like the hierarchy of ProGBN and HyperMiner, the regularization weight of ECRTM, and the graph neural networks of GINopic.
> 1. BERTopic has hyperparameters to set its clustering and dimension reduction modules, like the number of neighbors and components of UMAP; the min cluster size, min samples, metrics of HDBSCAN.
>
>
> __We explain the fewer hyperparameters of our model result from its extremely simplified structure.__
> We don't use the traditional VAE with complicated encoders and decoders or the complex dimension reduction and clustering modules as BERTopic.
> Our model only includes three kinds of embeddings and the embedding transport plans between them.
>
> Thank you for the comment. We've added more discussions to the paper.
>
> __Q7: cluster the word embeddings__
>
> We clarify __our method already works as clustering the word embeddings.__
> Figure_2(c) illustrates we can view our ETP as a clustering process: topic embeddings as cluster centers, word embeddings as samples, and the semantic relations between them as cluster assignments.
> We refine these assignments during learning through objective Eq(8).
>
> __Q8: take longer time to load high-resolution figures__
>
> Thank you for your comment. We've updated the paper with compressed figures.

---

> > ### Comment · Reviewer_YWcC · 2024-08-07
> > **Rebuttal Acknowledgment**
> >
> > Dear authors,
> >
> > thank you for your answers.
> > All of my questions have been adequately answered, or have been already answered within the original manuscript. Thank you for bringing that to my attention.
> >
> >
> > I have adapted my score accordingly
> > 5 -> 7

---

> > > ### Author Response · Authors · 2024-08-08
> > > **Response to Reviewer YWcC**
> > >
> > > Dear Reviewer YWcC,
> > >
> > > Thank you for your reply! We're glad that our responses have addressed your concerns.
> > >
> > > Thank you.

---

### Official Review · Reviewer_ceW2 · 2024-07-08

**Soundness:** 3
**Presentation:** 4
**Contribution:** 3
**Rating:** 7
**Confidence:** 3

**Summary:**

This paper introduces a fast, adaptive, stable, and transferable (FAST) topic modeling paradigm by using dual semantic-relation reconstruction (DSR) to model topic-document and topic-word relations. It enhances topic modeling by incorporating a embedding transport plan (ETP) method to address relation biases. Experimental results show FASTopic outperforms existing baselines.

**Strengths:**

1. This work introduces FASTopic by integrating the DSR paradigm, which is straightforward and provides a fresh aspect on handling semantic relations in topic modeling. Authors further propose ETP method to avoid relation bias issue.
2. Comprehensive experiments show FASTopic's superiority in effectiveness, efficiency, and adaptability, stability, and transferability.
3. The paper is well-written, and the code is available.

**Weaknesses:**

1. The method's performance may heavily rely on the document embedding model, which makes it harder to interpret the model results. It need to futher discuss that how specific changes in embeddings affect topic modeling.
2. The misplacement of tables and figures, such as Table~1, could potentially distract readers but do not detract significantly from the content's quality. But it does not matter.

**Questions:**

1. In Figure~2, is it intentional to design the variation in line thickness ($\pi_{11}$, $\pi_{12}$ and $\pi_{13}$) ?
2. Could the authors provide more detailed motivation and experimental validation for the choice of using pretrained document embedding model? How if train embedding model together?
3. What are the performance metrics (both training and inference) when FASTopic is run on a CPU environment?
4. Although it is claimed that FASTopic performs well without extensive hyperparameter tuning, the paper does not discuss experiments varying hyperparameters. Could you elaborate on how sensitive FASTopic is to changes in its hyperparameters?

**Limitations:**

No potential negative societal impacts.

---

> ### Author Rebuttal · Authors · 2024-08-07
>
> Thank you for your comments! We're glad that you believe our paper is well-written, our experiments are comprehensive, and our model is straightforward and fresh.
> We hope our responses can address your concerns and improve your rating.
>
>
> __Q1: how document embeddings affect the method__
>
> Thank you for the question. We clarify that __we have experimented with different document embeddings in Appendix_F (Table_10 and 11).__
> Generally, higher-quality document embeddings bring about better topic modeling performance.
> We explain that __relying on pretrained document embeddings has become prevalent in topic modeling now.__ This is because we can easily access plentiful high-quality document embeddings, such as Sentence-Transformers which provides document embeddings for 50+ languages.
>
>
> __Q2: why use pretrained document embeddings and what if train them together__
>
> We explain that __pretrained document embeddings contain abundant features, and using them is a common practice for topic modeling__, such as BERTopic, CombinedTM, and CTMneg. We don't need to retrain an embedding model, which saves us lots of effort.
> As mentioned in Section_3.4 Line_183, training the document embedding model together may lead to over-fitting problems, because our target datasets are regularly smaller than their pretraining datasets.
> This also greatly increases the training time and computational cost. Due to these reasons, the above early studies don't train document embeddings either.
>
>
>
> __Q3: running speed performance on CPU__
>
> Here we report the running time (seconds) of BERTopic and our model, including training (Train) and inference (Infer):
>
> |Model|20NG| | |NYT| | |WoS| | |NeurIPS| | |ACL| | |Wikitext-103| |
> |:----|:----|:----|:----|:----|:----|:----|:----|:----|:----|:----|:----|:----|:----|:----|:----|:----|:----|
> | |Train|Infer| |Train|Infer| |Train|Infer| |Train|Infer| |Train|Infer| |Train|Infer|
> |BERTopic|107.06|12.05| |120.70|9.55| |128.16|6.51| |208.23|89.89| |252.29|126.93| |592.62|228.08|
> |__FASTopic__|95.93|0.02| |112.96|0.02| |121.88|0.01| |177.58|0.01| |209.75|0.01| |552.70|0.04|
>
>
> We see that our model is slightly faster than BERTopic on training and has much shorter inference time.
> This is because BERTopic has to extract and compare the n-grams in documents for inference, while our model directly uses the fast matrix calculations in Eq(9) for inference.
> The other performance (like topic quality) is the same on CPU, since the training process remains unchanged.
>
> Thank you for the question. We've added these results to the paper.
>
>
> __Q4: hyperparameter sensitivity of the model__
>
> Here we report the results by varying the main hyperparameters $\varepsilon_{1}$ and $\varepsilon_{2}$ in Eq(5, 6):
>
> |$\varepsilon_{1}$|$C_V$|TD|Purity|NMI|
> |:----|:----|:----|:----|:----|
> |0.1|0.469|0.997|0.659|0.363|
> |0.2|0.470|1.000|0.682|0.364|
> |0.333|0.457|1.000|0.672|0.365|
> |0.5|0.432|1.000|0.681|0.372|
>
> |$\varepsilon_{2}$|$C_V$|TD|Purity|NMI|
> |:----|:----|:----|:----|:----|
> |0.1|0.424|1.000|0.669|0.353|
> |0.2|0.448|1.000|0.655|0.360|
> |0.333|0.435|1.000|0.677|0.368|
> |0.5|0.457|1.000|0.672|0.365|
>
>
> We see that the performance remains generally stable.
> Thank you for the question. We have added these results to the paper.
>
>
> __Q5: the line thickness in Figure 2__
>
> We vary the line thickness to indicate the values of semantic relations.
> Thank you for the comment. We have added this to the Figure 2 caption.
>
>
> __Q6: the positions of some tables__
>
> Thank you for the suggestion. We have updated their positions in the paper.

---

### Official Review · Reviewer_Py7Z · 2024-07-13

**Soundness:** 3
**Presentation:** 3
**Contribution:** 2
**Rating:** 6
**Confidence:** 4

**Summary:**

This paper proposes a fast, adaptive, stable, and transferable topic model, FASTopic. Instead of using the VAE or clustering method, it incorporates a new model structure named Dual Semantic-relation Reconstruction (DSR). DSR learns topics by directly optimizing the semantic relations among topics, documents, and words. The semantic relations are further regularized by an Embedding Transport Plan (ETP) method as an optimal transport problem. Experiments demonstrate the effectiveness of the proposed model.

**Strengths:**

1. The paper is well-written and easy to follow
2. The proposed DSR framework is simple and neat.
3. The experiments and ablation studies demonstrate the effectiveness and efficiency of the proposed FASTopic method.

**Weaknesses:**

1. [76,64] should be compared as baselines in the experiments since they both incorporate optimal transport objectives in their model as FASTopic does.
2.  As far as I know, the time complexity of solving the optimal transmission problem is very high. Is there any technique used in FASTopic to efficiently derive a solution? Furthermore, it would be great to theoretically analyze the time complexity of the current FASTopic framework.
3. Is it fair to compare FASTopic, which takes document embeddings as input extracted by sentence-BERT, with other baselines in the experiments?
4. As LLM has shown great performances in many NLP tasks, I believe it is necessary to include a discussion about the reasons and advantages of developing a topic model that is not LLM-based.

**Questions:**

See weaknesses.

**Limitations:**

Yes

---

> ### Author Rebuttal · Authors · 2024-08-07
>
> Thank you for your helpful feedback!  We're glad that you appreciate our well-written paper, neat method, and extensive experiments.
> We sincerely hope our responses can address your concerns and improve your rating.
>
>
> __Q1: comparison to earlier NSTM (2022) and WeTe (2023)__
>
> Thank you for your comment. We explain that __we've included the newest baselines__: ProGBN (ICML 2023), ECRTM (ICML 2023), and GINopic (NAACL 2024).
> Here we additionally report the results of NSTM and WeTe:
>
> |Model|20NG|||NYT|||WoS|||NeurIPS|||ACL|||Wikitext-103||
> |:----|:----|:----|:----|:----|:----|:----|:----|:----|:----|:----|:----|:----|:----|:----|:----|:----|:----|
> ||$C_V$|TD||$C_V$|TD||$C_V$|TD||$C_V$|TD||$C_V$|TD||$C_V$|TD|
> |NSTM|0.395|0.427||0.374|0.803||0.432|0.832||0.412|0.487||0.393|0.455||0.398|0.892|
> |WeTe|0.383|0.949||0.401|0.947||0.425|0.989||0.388|0.908||0.370|0.920||0.376|0.752|
> |__FASTopic__|__0.426__|__0.983__||__0.437__|__0.999__||__0.457__|__1.000__||__0.422__|__0.998__||__0.420__|__0.998__||__0.439__   |__0.992__|
>
> |Model|20NG| | |NYT| | |WoS| |
> |:----|:----|:----|:----|:----|:----|:----|:----|:----|
> | |Purity|NMI| |Purity|NMI| |Purity|NMI|
> |NSTM|0.354|0.356| |0.447|0.229| |0.476|0.262|
> |WeTe|0.268|0.304| |0.526|0.279| |0.555|0.349|
> |__FASTopic__|__0.577__|__0.525__| |__0.662__|__0.369__| |__0.672__|__0.365__|
>
> We see our model can surpass them too.
>
>
> __Q2: why solving the optimal transport is so fast__
>
> We explain that __we employ the fast Sinkhorn's algorithm to solve the optimal transport and our objective is much simpler.__
> As mentioned in Section_3.3 and Appendix_D, the Sinkhorn's algorithm is fast and suited to the execution of GPU (Cuturi_2013, Peyré_2019, Genevay_2019).
> Besides, as indicated in Section_3.4, __our objective is straightforward without complicated networks. It only invovles four parameters: topic and word embeddings and their weight vectors (Eq(8)).__
> Previous models, WeTe and ECRTM, also solve the optimal transport, but __their objectives rely on complicated encoders and decoders, which slows them down.__
> Thank you for the question. We've added more explanations in the paper.
>
>
>
> __Q3: is the comparison fair due to document embeddings?__
>
> We clarify that __our comparisons are fair since we've included the baselines using the same document embeddings.__
> BERTopic uses document embeddings for clustering; CombinedTM uses document embeddings as input features.
> These baselines use exactly the same document embeddings as our method for fair comparisons.
> Some other models originally cannot incorporate document embeddings but use pretrained word embeddings, like ProGBN, HyperMiner, and ECRTM.
>
>
>
>
> __Q4: discuss LLM-based topic models__
>
> LLM-based topic models are very promising such as TopicGPT, but we explain __they currently have two main limitations__:
> 1. __LLM-based topic models require more resources__. They need to input each document as prompts to LLMs. This is time-consuming and computationally intensive, especially when handling large-scale datasets.
> 2. __LLM-based topic models cannot produce precise distributions for topics and documents__. They can only use natural languages to describe topics and documents.
>
> As shown in the paper, our model has a fast running speed (Figure_1) and also provides precise distributions (Section_3.3).
> Thank you for the question. We have added more discussions in the related work.

---

### Official Review · Reviewer_FTng · 2024-07-13

**Soundness:** 3
**Presentation:** 3
**Contribution:** 3
**Rating:** 5
**Confidence:** 4

**Summary:**

The author found that existing methods (VAE-based or clustering) suffer from low efficiency, poor quality of topic words, and instability. To address these issues, this paper proposes a novel topic modeling paradigm called Dual Semantic-Relation Reconstruction (DSR) for efficient modeling of semantic relations among three types of embeddings: document, topic, and word embeddings. The author attributes the low quality and instability of previous methods to relation bias issues, leading to repetitive topics and inaccurate document-topic distributions. To tackle this, the paper introduces the Embedding Transport Plan (ETP) to regularize relations among the three embeddings. Combined, DSR and ETP form the proposed topic model, FASTopic, which is evaluated on six benchmark datasets, showing encouraging performance.

**Strengths:**

1. The paper is clearly presented and easy to follow, with an explanatory and easy-to-understand mathematical formulation.
2. The proposed DSR objective is effective, significantly reducing training time compared to VAE-based methods such as ECRTM and CombinedTM.
3. The experimental evaluation is extensive, showing that FASTopic consistently outperforms multiple strong baselines on topic coherence and topic diversity, while also demonstrating advantages in terms of running speed and transferability.
4. The paper demonstrates the model's robustness under multiple numbers of topics (K=75, 100, ..., 200).

**Weaknesses:**

1. Tables 6, 8, and 9 present the ablation study of FASTopic using ETP and parameterized softmax. ETP, compared to the ECR used in the previous VAE-based method ECRTM, adds semantic relations between topics and documents. The paper lacks proof the dual transmission effectiveness of ETP by not comparing it with ECR (which only includes semantic relations between topics and words) and another case that only includes semantic relations between documents and topics.
2. I am concerned that while the DSR training method improves efficiency compared to VAE-based methods, it may make it difficult to build meaningful relations between topic embeddings and document embeddings for some long-text corpora due to its singular objective.
3. It would be nice to showcase a comparison of topic words or transferability between FASTopic and ECRTM to demonstrate that FASTopic offers improvements in multiple dimensions, not just speed.

**Questions:**

1. In Line 76, the author thinks that methods like Topicgpt, which uses LLM to describe topics, deviate from the original LDA setting. Can the author provide a more detailed explanation in this section?
2. In Line 132, it is mentioned that some studies think that repetitive topics and less accurate doc-topic distributions are due to a large number of topics (K being set too high). The paper, however, attributes these issues mainly to relation bias. Does a large number of topics also affect FASTopic in the same way?

**Limitations:**

1. The proposed method relies on embedding transport semantic relations and may be limited by the max input length of pre-trained document embedding models.
2. See Weakness 3.

---

> ### Author Rebuttal · Authors · 2024-08-07
>
> Thank you for your feedback! We're happy that you appreciate our clear writing, effective method, and extensive experiments.
> We sincerely hope our responses can address your concerns and improve your rating.
>
>
> __Q1: difference between ETP and the ECR of ECRTM (Wu_2023)__
>
> We clarify that __the ECR of ECRTM is NOT an alternative to our ETP for ablation study.__
> The ECR (embedding clustering regularization) of ECRTM only regularizes topic and word embeddings and does __NOT__ model the semantic relations between them as topic-word distributions. ECRTM follows the traditional VAE to model topic-word and doc-topic distributions for topic modeling.
> Instead of VAE, our method uses ETP to model the semantic relations as topic-word and doc-topic distributions for topic modeling.
> Thus, the previous ECR does not model the semantic relations as our ETP, and it is NOT an alternative to our ETP for ablation study.
> Besides, only the semantic relations between documents and topics or only between topics and words cannot do topic modeling, because we need both of them for reconstruction in Eq(1).
>
> Thank you for the question. We've added more clarifications in the related work.
>
>
>
> __Q2: performance on long-text corpora__
>
> We clarify that __we have experimented on long-context corpora, NeurIPS, ACL, and Wikitext-103.__
> As reported in Appendix_B Table_7, they are long academic publications or Wikipedia articles, ranging from __1k to 2k words__. We have shown that our model works well on these long-context corpora in Section_4.2, 4.3, and 4.4.
>
>
> __Q3: showcase comparisons to ECRTM__
>
> We clarify that __we have showcased the comparisons to ECRTM on topic words and doc-topic distributions (Section_4.2), running speed (Section_4.4), transferability ( Section_4.5), and adaptivity (Section_4.6).__
>
> Here we illustrate some cases of topic words.
>
> |ECRTM|
> |:----|
> |#14: boeing software microsoft customer trains airline hardware updates airlines consoles|
> |#50: typhoon carrier boeing aircraft airlines carriers airline philippines pilots japanese|
>
> |FASTopic|
> |:----|
> |#18: mario computer software gamer eurogamer graphical puzzles informer consoles microsoft|
> |#22: trains train airport passengers passenger stations cars airline ride roller|
> |#43: tropical hurricane cyclone storm winds rainfall depression flooding intensity mph|
>
>
> We see ECRTM mixes the topics of airlines, software, and typhoon.
> In contrast, FASTopic produces more separated topics about them respectively.
>
>
> __Q4: Differences between TopicGPT and LDA__
>
> We explain that __their main difference lies in how they define topics and understand documents.__
> LDA defines a topic as a word distribution, for example, a distribution [0.1, 0.2, 0.1 ...] over words [farmer, products, agricultural, ...].
> Differently, TopicGPT defines a topic as a natural language description, like *Agriculture: Discusses policies relating to agricultural practices and products...*.
> Besides, LDA infers distributions over topics to understand documents, while TopicGPT classifies a document with topics as label space by prompting.
> TopicGPT reaches higher interpretability, but cannot give precise distributions for downstream tasks.
>
> Thank you for the question. We have updated the paper with more explanations.
>
>
>
> __Q5: does a large number of topics affect FASTopic?__
>
> We clarify that __we've reported in Table 4 and 5 the results under large numbers of topics (K=75, 100, ... 200).__ Our model remains high-performance with large numbers of topics, especially on topic diversity.
>
> We clarify that in Line_132, we intend to show using parameterized softmax leads to the relation bias issue even with a small number of topics, as supported by Table_9. This motivates us to propose the new ETP method.

---

> ### Author Response · Authors · 2024-08-14
> **Looking forward to your further feedback!**
>
> Dear Reviewer FTng,
>
> Thank you sincerely for your previous helpful reviews!
> We mention that we've submitted our responses, including __clarifications on our differences from ECRTM and our performance on long-text corpora and large numbers of topics.__
> We hope our responses can address your concerns. We are looking forward to your insightful feedback!
>
> Thank you.
>
> Best,
> Authors

---

### Decision · Program_Chairs · 2024-09-25

**Decision:**

Accept (poster)

**Comment:**

Summary
============
This paper parameterizes the creation of topic embeddings from semantic relations over word embeddings and semantic relations mapping fixed document embeddings to topic embeddings.

Justification
============
The method is fast and does well on an impressive, comprehensive set of *automated* metrics.  I like the approach, and the reviewers do a good job of outlining the strength of the approach.

Where I think the paper still falls short is:
* The paper does not compare against any conventional topic model approaches, which can be quite fast.  E.g., while Arora et al. is cited, they do not compare against this approach.  Stochastic variational inference could also be compared against, and it's unclear if they compare against Yao et al's fast Gibbs Sampling.
* There are now human evaluations.  While Hoyle et al. is cited, they do not follow the advice of the paper to vet automatic topic evaluations with human evaluations, particularly for neural models.
* The exposition could be clearer; I would have liked to have seen a little more intuition for describing the semantic relations (e.g., with a toy example of generating the representations).

Ultimately, I think this could be a *great* paper with another round of revision, but it isn't quite there yet.